# FROM VICIOUS TO VIRTUOUS CYCLES: SYNERGISTIC REPRESENTATION LEARNING FOR UNSUPERVISED VIDEO OBJECT-CENTRIC LEARNING

**Hyun Seok Seong**[*] **WonJun Moon**[*] **Jae-Pil Heo**[†]
Sungkyunkwan University
{gustjrdl95, wjun0830, jaepilheo}@skku.edu

## ABSTRACT

Unsupervised object-centric learning models, particularly slot-based architectures, have shown great promise in decomposing complex scenes. However, their reliance on reconstruction-based training creates a fundamental conflict between the sharp, high-frequency attention maps of the encoder and the spatially consistent but blurry reconstruction maps of the decoder. We identify that this discrepancy gives rise to a vicious cycle: the noisy feature map from the encoder forces the decoder to average over possibilities and produce even blurrier outputs, while the gradient computed from blurry reconstruction maps lacks high-frequency details necessary to supervise encoder features. To break this cycle, we introduce Synergistic Representation Learning (SRL) that establishes a virtuous cycle where the encoder and decoder mutually refine one another. SRL leverages the encoder's sharpness to deblur the semantic boundary within the decoder output, while exploiting the decoder's spatial consistency to denoise the encoder's features. This mutual refinement process is stabilized by a warm-up phase with a slot regularization objective that initially allocates distinct entities per slot. By bridging the representational gap between the encoder and decoder, SRL achieves state-of-the-art results on video object-centric learning benchmarks. Codes are available at github.com/hynnsk/SRL.

## 1 INTRODUCTION

Object-centric representation learning aims to decompose complex scenes into a set of disentangled object representations, a critical capability for robust video understanding (Xu et al., 2024). Among prevailing approaches, slot-based models (Locatello et al., 2020; Manasyan et al., 2025; Zadaianchuk et al., 2023; Kipf et al., 2021; Elsayed et al., 2022) have demonstrated significant promise in learning to group pixels into meaningful object-level slots in an unsupervised manner. These models typically operate by encoding a video into a feature map, which is then parsed by an attention mechanism into a fixed number of latent slots. The quality of these slots is subsequently evaluated by a decoder that attempts to reconstruct the original input from them, using a reconstruction loss like Mean Squared Error (MSE) as the primary training signal. This reconstruction-based supervision is vital, as it circumvents the need for manual annotations and provides a workable objective in a purely unsupervised setting where direct supervision on feature grouping is noisy and difficult to formulate.

However, we identify a fundamental discrepancy inherent in this widely adopted training paradigm. The learning process relies on two distinct spatial maps that are unfortunately misaligned in their characteristics: (1) the attention maps generated by the slot attention, and (2) the decoded output maps produced by the reconstruction decoder. The attention maps, derived from pixel-wise feature similarities, are inherently sharp and granular, but also susceptible to high-frequency noise. In contrast, the decoded output maps, typically generated by passing the flattened slots through an MLP decoder, tend to be blurry and spatially smooth. This blurring effect is an artifact of the autoencoder's architecture (e.g., Slot Attention) and the smoothing nature of the MSE loss (Mustafa et al., 2022; Zhao et al., 2016), which leads to perceptual compression (Rombach et al., 2022).

---

[*]Equal Contribution
[†]Corresponding Author

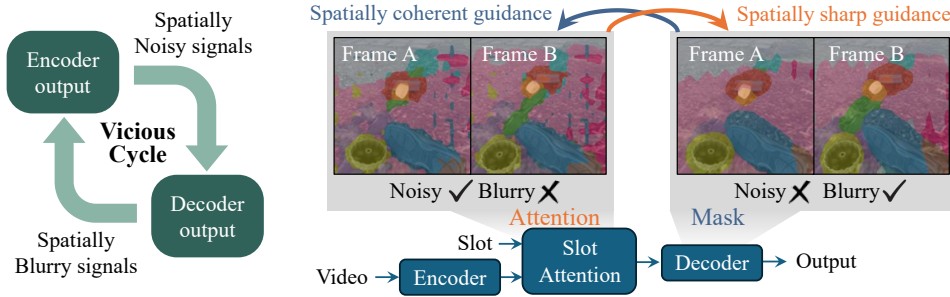

(a) Vicious cycle        (b) From vicious to virtuous cycle via synergistic learning

Figure 1: (a) Vicious cycle in video object-centric learning. Noisy inputs from the encoder render the decoder's reconstruction task ill-posed, reinforcing its tendency to produce blurry, low-frequency outputs. In turn, the corrupted gradient from these blurry outputs lacks the high-frequency detail required to refine the encoder's sharp but noisy features. (b) Virtuous cycle of synergistic representation learning. Our framework transforms this conflict into collaboration. We leverage the encoder's sharp attention maps to deblur the decoder output while denoising the encoder features with the decoder's spatially coherent masks.

This discrepancy incurs a vicious feedback loop that fundamentally constrains the learning process, as shown in Fig. 1. On one hand, the encoder, while leveraging sharp DINO-v2 features (Oquab et al., 2024), produces noisy groupings by incorrectly associating spatially distant patches (Yang et al., 2024). When the decoder receives these noisy slot representations, its reconstruction task becomes ill-posed. To minimize the MSE penalty under this uncertainty, the decoder's safest strategy is to average over the possibilities, which further reinforces its own tendency to produce blurry outputs by a biased optimization toward recovering low-frequency content. On the other hand, such a decoder provides a corrupted, low-frequency learning signal to the rest of the model. The gradients flowing back to the encoder lack the precise high-frequency details necessary to supervise the learning of sharp encoder features.

To break this vicious cycle and establish a virtuous one, we introduce Synergistic Representation Learning, a novel framework where the two spatial maps synergistically refine one another through purpose-built objectives. First, we tackle the decoder's blurriness by leveraging the encoder's sharp, albeit noisy, attention map as a guide. We introduce a ternary contrastive objective for deblurring that strategically partitions patches into three tiers: the anchor itself, other patches grouped with the anchor by the encoder's sharp attention map, and all other patches. A ranking loss then enforces this objective, compelling the decoder to resolve ambiguities at the object boundary where its blurry grouping conflicts with the encoder's sharp prior. On the other hand, we leverage the decoder's more spatially coherent representation to provide a training signal to denoise the encoder's noisy representations. Specifically, we exploit another ternary contrastive objective to use the decoder's consistent masks to enforce spatial consistency within encoded feature maps, pushing spuriously grouped, distant patches apart in the feature space. This entire refinement process is built upon a robust warm-up phase that employs a slot regularization loss. This prevents the initial slot collapse by identifying and resetting redundant slots, ensuring a meaningful foundation for the subsequent decoder deblurring and encoder denoising processes. Through this co-evolving optimization process, our method successfully bridges the gap between the noisy encoder and the blurry decoder, resulting in significantly sharper object segmentation and more robust representations.

## 2 RELATED WORK

### 2.1 OBJECT-CENTRIC REPRESENTATION LEARNING

The goal of object-centric learning is to decompose complex scenes into a set of discrete object representations without explicit supervision (Kirilenko et al., 2024; Singh et al., 2021; Kosiorek et al., 2018; Burgess et al., 2019; Lin et al., 2020). A significant breakthrough in this area is Slot Attention (Locatello et al., 2020), which employs an iterative, competitive attention mechanism to bind slots to different objects in an image. Each slot, initialized randomly, refines its representation over several iterations by competing for evidence from the input features, effectively performing a soft, differentiable version of clustering.

This paradigm was successfully extended to the temporal domain for video processing. Earlier works like SAVi (Kipf et al., 2021; Elsayed et al., 2022) and STEVE (Singh et al., 2022) maintain temporal consistency by propagating slot representations from one frame to the next, enabling robust unsupervised object tracking and decomposition in dynamic scenes. Subsequently, Videosaur (Zadaianchuk et al., 2023) proposed a self-supervised task to predict patch motion, and more recently, SlotContrast (Manasyan et al., 2025) introduced slot-level contrastive learning between slots of successive frames to enhance temporal consistency.

Other popular streams include reducing the redundancy in slot representations and self-distillation. To reduce the redundancy between slots, SOLV (Aydemir et al., 2023) merges slots via a non-differentiable agglomerative clustering procedure, while MetaSlot (Liu et al., 2025) addresses redundancy by using a codebook to prune duplicated slots. While these approaches are effective, they rely on explicit redundancy detection, which evolves relatively slowly during training. Similarly, we employ a slot regularization objective to mitigate redundancy. However, a key difference is that our regularization is aggressively applied only during the initial training iterations. This allows the encoder-decoder architecture to achieve stable representations early in training, establishing a robust foundation for the subsequent learning process.

The other stream of research focuses on self-distillation (Kakogeorgiou et al., 2024; Zhao et al., 2025). For instance, SPOT (Kakogeorgiou et al., 2024) distills decoder signals into encoder attention, while DIAS (Zhao et al., 2025) transfers later-iteration attention to earlier steps. While effective in certain settings, these methods directly imitate teacher attention signals without explicitly addressing the noise inherent in the teacher's knowledge. In contrast, our synergistic representation learning aims to leverage only the complementary strengths of encoder and decoder representations. We mitigate the impact of noisy signals by stratifying mutual signals into intermediate levels, rather than enforcing strong positive or negative constraints.

## 2.2 CONTRASTIVE REPRESENTATION LEARNING

Contrastive representation learning (Chen et al., 2020; He et al., 2020; Grill et al., 2020; Caron et al., 2020; Moon et al., 2025) has emerged as a powerful paradigm for learning discriminative embeddings by encouraging semantically similar samples to be mapped closer together while pushing apart dissimilar ones. This objective is typically instantiated through the InfoNCE loss (Oord et al., 2018), which enforces such pairwise alignment in the embedding space. Building upon this foundation, supervised extensions (Khosla et al., 2020; Kang et al., 2020) have demonstrated the effectiveness of leveraging multiple positives per anchor, showing that clustering semantically consistent samples enhances representation quality.

Extending this idea to the unsupervised setting, several works have explored strategies for mining semantically similar samples, such as selecting top-$K$ nearest neighbors as positives (Dwibedi et al., 2021; Seong et al., 2023; 2024). These approaches demonstrate that expanding the set of positives beyond simple augmentations leads to more robust feature clustering. Parallel to positive mining, another line of work has emphasized the importance of hard negative selection, showing that the quality of hard negatives is crucial for effective contrastive learning (Robinson et al., 2021; Kalantidis et al., 2020).

Inspired by these insights, we introduce a contrastive framework that leverages the complementary conflict between the encoder's sharp but noisy features and the decoder's coherent but blurry masks. By defining a ternary structure of patch relationships, we weaponize this discrepancy: the encoder's sharpness provides a deblurring signal for the decoder, while the decoder's coherence provides a denoising signal for the encoder. This cross-source hard negative mining compels each module to overcome its weaknesses, resulting in a synergistic refinement of object-centric representations.

## 3 METHOD

We address a vicious cycle in video object-centric learning caused by a representational conflict between the slot attention maps and decoded output masks after reconstruction. This conflict recursively inhibits optimization, preventing the learning of clean object representations. To break this, we introduce Synergistic Representation Learning (SRL), which combines mutual refinement process via contrastive losses with a slot-regularization warm-up phase. As presented in Fig. 2, SRL initially

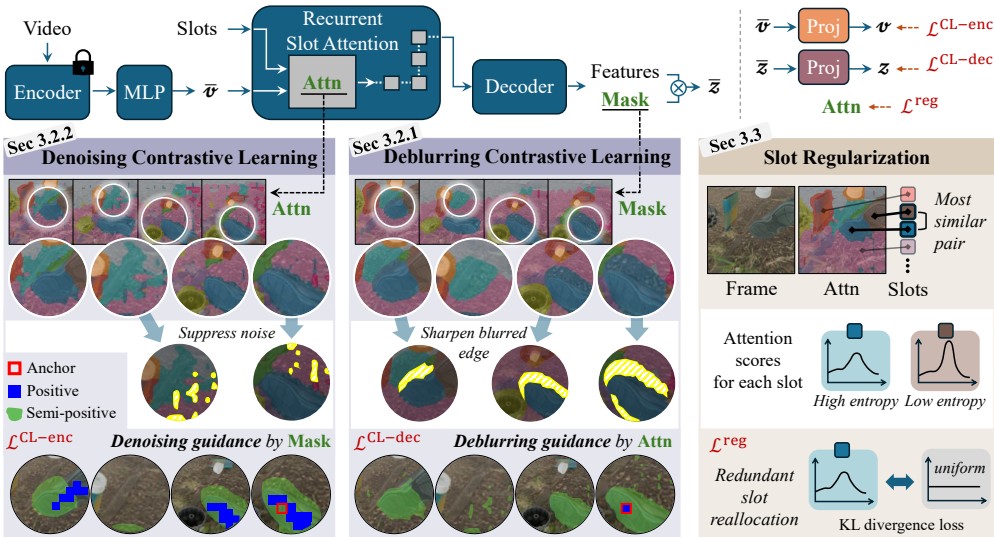

Figure 2: Overview of Synergistic Representation learning. The typical pipeline (top) suffers from a conflict between the encoder's sharp but noisy features ($\bar{v}$) and the decoder's spatially coherent but blurry features ($\bar{z}$). Our framework breaks this cycle by forcing the two modules to synergistically refine one another: (1) Deblurring path: Encoder's sharp attention map is used to refine the blurry decoded features and (2) Denoising path: Decoder's coherent masks provide a robust signal to denoise the encoder's noisy features. Finally, slot regularization during warm-up establishes a solid foundation for this process by ensuring diverse slot specialization.

uses slot regularization to promote robust and non-redundant slot specialization, and then establishes a virtuous cycle in which the slot attention maps and decoded masks iteratively refine each other through contrastive learning.

## 3.1 VICIOUS CYCLE OF ENCODER-DECODER DISCREPANCY

The standard paradigm for training video slot attention models relies on a reconstruction objective (Zadaianchuk et al., 2023; Locatello et al., 2020), creating an inherent conflict between the encoder's grouping mechanism and the decoder's learning signal. This conflict establishes a vicious cycle that hinders optimization.

**Noisy Encoder Reinforces the Decoder's Blurriness.** The encoder's features, which are passed to the Slot Attention, are derived from projecting the DINO-v2 (Oquab et al., 2024; Zhou et al., 2022) features, which are designed to be discriminative at a fine-grained level. This results in representations that are sharp but also susceptible to noise, such as incorrectly grouping spatially distant patches (Yang et al., 2024). Subsequently, when the decoder is conditioned on a limited number of noisy latent slots, the reconstruction task becomes ill-posed. To minimize the MSE loss under this uncertainty, the decoder's safest strategy is to average over the possibilities, further reinforcing its tendency to produce blurry, over-smoothed outputs.

**Blurry Decoder Corrupts the Encoder's Learning Signal.** Conversely, the decoder is trained with a pixel-wise objective like MSE, which inherently acts as a low-pass filter. It incentivizes the model to predict the conditional expectation of pixel values, resulting in reconstructions that are spatially smooth but suffer from blurry object boundaries and a loss of high-frequency detail. The gradient signal that trains the entire model, including the encoder, originates from this blurry output. Consequently, the encoder receives a signal that fails to provide the precise guidance needed to learn sharp object boundaries and refine the noisy patch representations.

## 3.2 FROM VICIOUS TO VIRTUOUS CYCLES VIA SYNERGISTIC REPRESENTATION LEARNING

We aim to break this vicious cycle with SRL, equipped with two purpose-built contrastive objectives that force the encoder and decoder to refine their representations mutually. Our representation learning begins after a warmup stage, allowing the model to leverage two distinct spatial maps: the slot attention

map ($\textbf{Attn} \in \mathbb{R}^{S \times T \times N}$) that is sharp yet noisy, and the decoded mask ($\textbf{Mask} \in \mathbb{R}^{S \times T \times N}$) that is spatially consistent yet blurry. $S, T$, and $N$ denote the number of slots, frames, and spatial patches. From these probabilistic maps, we derive hard pseudo-semantic labels. For a given frame $t$ and spatial patch $i$, pseudo-semantic labels are defined as:

$$l_{t,i}^{\textbf{Attn}} = \underset{s \in \{1,\dots,S\}}{\arg\max} \, \textbf{Attn}_{s,t,i} \quad ; \quad l_{t,i}^{\textbf{Mask}} = \underset{s \in \{1,\dots,S\}}{\arg\max} \, \textbf{Mask}_{s,t,i}. \tag{1}$$

The label $l^{\textbf{Attn}}$ represents the index of the slot that gives the highest attention score to a specific patch, while $l^{\textbf{Mask}}$ represents the index of the slot with the highest value in the decoded mask for that patch.

### 3.2.1 DEBLURRING CONTRASTIVE LEARNING: REFINING THE DECODER VIA ENCODER SHARPNESS

To counteract the decoder's blurry object boundaries, we introduce a contrastive objective, $\mathcal{L}_{\text{CL-dec}}$. Our key insight is to weaponize the representational discrepancy between the modules, using the encoder's sharp, attention-derived groupings to mine hard negatives that explicitly penalize blurriness in the decoder's representation space. To align our contrastive objective with the reconstruction loss, we formulate the contrastive task as aligning the decoded features with their corresponding features from the backbone encoder, which also serves as the target guidance for the reconstruction loss. Following typical conventions (Chen et al., 2020; Khosla et al., 2020), we project both feature sets into another embedding space, where a contrastive loss is then calculated.

To guide the decoder toward producing crisp, high-fidelity object boundaries, we organize ternary sets of patches for each anchor patch that explicitly preserves the reliability ordering of patch relationships. For each anchor patch, identified by its spatio-temporal index $(t, i)$ where $t$ is the frame and $i$ is the spatial location, we partition the universal set of all patch indices $\mathcal{U}$. This universal set $\mathcal{U}$ contains all possible index pairs $(t', j)$ across all frames $t' \in \{1, \dots, T\}$ and all flattened spatial locations $j \in \{1, \dots, N\}$. The set $\mathcal{U}$ is partitioned into three disjoint subsets relative to the anchor $(t, i)$: (1) the positive set $\mathcal{P}_{t,i}^{\text{dec}}$, (2) the semi-positive set $\mathcal{Q}_{t,i}^{\text{dec}}$, and (3) the negative set $\mathcal{N}_{t,i}^{\text{dec}}$:

$$\mathcal{U} = \mathcal{P}_{t,i}^{\text{dec}} \cup \mathcal{Q}_{t,i}^{\text{dec}} \cup \mathcal{N}_{t,i}^{\text{dec}}. \tag{2}$$

First, the positive set is defined as the anchor itself ($\mathcal{P}_{t,i}^{\text{dec}} = \{(t, i)\}$). This anchors the objective to self-reconstruction with the aim of semantic deblurring. Yet, naïvely treating all other patches as negatives may corrupt the semantic boundaries by erroneously pushing away patches belonging to the same object. Therefore, we further distinguish semantically similar and organize the semi-positive set, using the encoder's sharp, attention-derived labels ($l^{\textbf{Attn}}$) as:

$$\mathcal{Q}_{t,i}^{\text{dec}} = \{ \, (t', j) \mid l_{t',j}^{\textbf{Attn}} = l_{t,i}^{\textbf{Attn}} \, \}, \tag{3}$$

and its purpose is to further guide the decoder to learn the structural prior from the encoder. Lastly, the negative set $\mathcal{N}_{t,i}^{\text{dec}}$ is defined as the complement, containing all semantically distinct patches. Crucially, this set also includes patches from blurred object boundaries, which the decoder might incorrectly associate with the anchor's object. By including these ambiguous boundary patches in the negative set, we create a contrastive pressure that further compels the decoder to learn a more deblurred representation of the object.

To enforce this hierarchical structure, we adopt a ranking contrastive loss (Hoffmann et al., 2022) that operates on two levels. This preserves the reliability ordering by ensuring the anchor is pulled more strongly to its own ground-truth than to its semi-positives, and to its semi-positives more strongly than to its negatives. Formally, our decoder contrastive loss $\mathcal{L}^{\text{CL-dec}}$ is expressed as:

$$\mathcal{L}_{t,i}^{\text{CL-dec}} = -\log \frac{\exp(\boldsymbol{z}_{t,i} \cdot \boldsymbol{y}_{t,i}/\tau)}{\displaystyle\sum_{n \in \mathcal{Q}_{t,i}^{\text{dec}} \cup \mathcal{N}_{t,i}^{\text{dec}}} \exp(\boldsymbol{z}_{t,i} \cdot \boldsymbol{y}_n/\tau)} - \frac{1}{|\mathcal{Q}_{t,i}^{\text{dec}}|} \sum_{q \in \mathcal{Q}_{t,i}^{\text{dec}}} \log \frac{\exp(\boldsymbol{z}_{t,i} \cdot \boldsymbol{y}_q/\tau)}{\displaystyle\sum_{n \in \mathcal{N}_{t,i}^{\text{dec}}} \exp(\boldsymbol{z}_{t,i} \cdot \boldsymbol{y}_n/\tau)}, \tag{4}$$

where $\cdot$ denotes cosine similarity, $\tau$ is a temperature parameter. $\boldsymbol{z} \in \mathbb{R}^{T \times N \times C}$ and $\boldsymbol{y} \in \mathbb{R}^{T \times N \times C}$ are projected vectors of decoder and backbone features, respectively, where $C$ is the channel dimension. This structured signal directly counteracts the low-pass filtering effect of the MSE loss, guiding the decoder to produce crisp object boundaries.

### 3.2.2 DENOISING CONTRASTIVE LEARNING: REFINING THE ENCODER VIA DECODER COHERENCE

Conversely, while the encoder's features inherited from a powerful backbone (e.g., DINO-v2 (Oquab et al., 2024)) are spatially sharp, they are susceptible to assigning high similarity to spurious, far-flung patches (i.e., noise). To resolve this, we leverage the decoder's spatially coherent masks to denoise the encoded MLP features $\bar{v}$. We instantiate this with a second ternary contrastive objective, $\mathcal{L}^{\text{CL-enc}}$.

The objective's structure is similar to that of $\mathcal{L}^{\text{CL-dec}}$, partitioning patches into positive, semi-positive, and negative sets. However, the sets are defined differently to serve the specific goal of denoising rather than sharpening. To illustrate, for a given anchor patch $(t, i)$, the positive set $\mathcal{P}^{\text{enc}}_{t,i}$ is defined by leveraging the semantic similarity within the backbone. It comprises the anchor's Top-$K$ nearest neighbors in the DINO-v2 embedding space, sampled from all $T$ frames of the video. This anchors the representation to the strongest semantic signals provided by the backbone, grouping similar patches. The semi-positive set $\mathcal{Q}^{\text{enc}}_{t,i}$ is gathered to enforce spatial coherence. It is defined using the coarse (blurred) but contiguous object masks generated by the decoder as follows:

$$\mathcal{Q}^{\text{enc}}_{t,i} = \{\, (t', j) \mid (l^{\textbf{Mask}}_{t',j} = l^{\textbf{Mask}}_{t,i}) \,\}, \tag{5}$$

where all patches that share the same decoder-derived label ($l^{\textbf{Mask}}$) form this set. The negative set $\mathcal{N}^{\text{enc}}_{t,i}$ is defined as the complement.

Then, we apply the same ranking loss as in Eq. 4, except that the projected decoder $\hat{y}$ and backbone features $y$ are both replaced by the projected MLP features $v$. Then, the objective is expressed as:

$$
\begin{aligned}
\mathcal{L}^{\text{CL-enc}}_{t,i} = &- \frac{1}{|\mathcal{P}^{\text{enc}}_{t,i}|} \sum_{p \in \mathcal{P}^{\text{enc}}_{t,i}} \log \frac{\exp(v_{t,i} \cdot v_p / \tau)}{\sum_{n \in \mathcal{Q}^{\text{enc}}_{t,i} \cup \mathcal{N}^{\text{enc}}_{t,i}} \exp(v_{t,i} \cdot v_n / \tau)} \\
&- \frac{1}{|\mathcal{Q}^{\text{enc}}_{t,i}|} \sum_{p \in \mathcal{Q}^{\text{enc}}_{t,i}} \log \frac{\exp(v_{t,i} \cdot v_p / \tau)}{\sum_{n \in \mathcal{N}^{\text{enc}}_{t,i}} \exp(v_{t,i} \cdot v_n / \tau)}.
\end{aligned}
\tag{6}
$$

This formulation uses the larger positive set to ensure features of the same class cluster together, while the decoder-derived semi-positive set tightens this cluster around a spatially coherent instance, effectively exposing the noise patches in the negative set.

## 3.3 SLOT REGULARIZATION FOR REDUNDANCY REDUCTION

A reliable initial assignment of slots to objects is a critical prerequisite for our mutual refinement process. Our contrastive objectives operate at a fine-grained level and are intended to calibrate slot representations after the slots have already captured the coarse semantics of distinct objects. However, in practice, objects are often fragmented into multiple slots when the model aggressively minimizes reconstruction error, leading to several redundant slots covering the same object region. In such cases, these spatially overlapping slots may continue to cooperate and further fragment the object instead of consolidating it. To prevent this degenerate behavior, we introduce a slot regularization objective.

This regularization identifies and penalizes $M$ most redundant slots, iteratively performing the following steps $M$ times. First, the model identifies the most similar slot pair, $(\hat{i}, \hat{j})$, by finding the pair of indices that maximizes the cosine similarity between their final representations at frame $T$:

$$(\hat{i}, \hat{j}) = \underset{1 \leq i < j \leq S}{\arg\max} \; (s_{T,i} \cdot s_{T,j}). \tag{7}$$

For the identified pair, it assesses which slot is less specialized to specific semantics. Specialization is measured by mean KL divergence between attention maps for a given slot $m$, denoted as $\textbf{Attn}_m$, and a uniform distribution $\textbf{U}$ across all $T$ frames. The slot with the lower score is selected for regularization:

$$m^{\text{low}} = \underset{m \in \{\hat{i}, \hat{j}\}}{\arg\min} \frac{1}{T} \sum_{t=1}^{T} D^{\text{KL}} \left( \textbf{Attn}_{m,t} | \textbf{U} \right). \tag{8}$$

The index of the chosen slot, $m^{\text{low}}$, is then added to a set of penalized indices, $\mathcal{M}^{pen}$. This slot is subsequently excluded from consideration in the remaining selection steps.

Table 1: Experimental results. Results are averaged across 3 runs. † is our reproduced version.

| Method | MOVi-C | | MOVi-E | | YouTube-VIS | |
|---|---|---|---|---|---|---|
| | FG-ARI↑ | mBO↑ | FG-ARI↑ | mBO↑ | FG-ARI↑ | mBO↑ |
| SAVi (Kipf et al., 2021) | 22.2 | 13.6 | 42.8 | 16.0 | - | - |
| STEVE (Singh et al., 2022) | 36.1 | 26.5 | 50.6 | 26.6 | 15.0 | 19.1 |
| VideoSAUR (Zadaianchuk et al., 2023) | 64.8 | **38.9** | 73.9 | **35.6** | 28.9 | 26.3 |
| VideoSAURv2 (Manasyan et al., 2025) | – | – | 77.1 | 34.4 | 31.2 | 29.7 |
| SlotContrast (Manasyan et al., 2025) | 69.3 | 32.7 | 82.9 | 29.2 | 38.0 | 33.7 |
| SlotContrast† (Manasyan et al., 2025) | 70.4 | 31.7 | 80.9 | 28.2 | 36.2 | 32.9 |
| SRL (Ours) | **74.3** | 34.5 | **81.9** | 29.3 | **42.9** | **35.6** |

After this iterative process populates the set $\mathcal{M}^{\text{pen}}$ with $M$ slot indices, we regularize the corresponding slots. The attention distribution of each penalized slot is encouraged to align with a uniform distribution $\mathbf{U}$ via the following KL divergence loss:

$$\mathcal{L}^{\text{reg}} = \frac{1}{MT} \sum_{m \in \mathcal{M}_{\text{pen}}} \sum_{t=1}^{T} D^{\text{KL}} \left( \mathbf{Attn}_{m,t} | \mathbf{U} \right), \qquad (9)$$

where $\mathbf{Attn}_{m,t}$ is the attention map of the specific penalized slot with index $m$ at frame $t$. This warm-up regularization encourages redundant, less-specialized slots to abandon their overlap and instead discover unexplained regions of the scene, thereby laying a strong foundation for the subsequent mutual refinement. See Appendix A.2 for a visual illustration of the warm-up effect.

### 3.4 STAGED TRAINING FRAMEWORK

Along with our proposed objectives, we adopt the baseline loss, $\mathcal{L}^{\text{base}}$, from SlotContrast (Manasyan et al., 2025), which is composed of the MSE reconstruction loss and the slot-level contrastive loss. The baseline objectives are applied throughout the training, while our proposed losses are progressively activated as the model's internal representations become more structured.

Specifically, training proceeds in three stages: (i) Slot specialization (0-10%), (ii) Slot stabilization (10-20%), and (iii) Contrastive refinement (20-100%). During the slot specialization stage, we introduce the regularization loss $\mathcal{L}^{\text{reg}}$, which encourages early semantic differentiation among slots. Then, we train solely with $\mathcal{L}^{\text{base}}$ to consolidate stable slot representations. Finally, we activate our core contribution, $\mathcal{L}^{\text{CL}}$, as both the encoder and decoder representations are sufficiently meaningful for their discrepancy to serve as a rich, structured learning signal at this stage. The overall objective is as:

$$\mathcal{L} = \mathcal{L}^{\text{base}} + \mathcal{L}^{\text{stage}} \;\; ; \;\; \mathcal{L}^{\text{stage}} = \begin{cases} \lambda^{\text{reg}} \mathcal{L}^{\text{reg}}, & \text{if } \eta < 0.1, \\ 0, & \text{if } 0.1 \leq \eta < 0.2, \\ \lambda^{\text{CL}} \mathcal{L}^{\text{CL}}, & \text{if } \eta \geq 0.2, \end{cases} \qquad (10)$$

where $\lambda^{\text{reg}}$ and $\lambda^{\text{CL}}$ are loss coefficients, $\eta$ is a ratio of training progress, and $\mathcal{L}^{\text{CL}} = \mathcal{L}^{\text{CL-enc}} + \mathcal{L}^{\text{CL-dec}}$.

## 4 EXPERIMENT

### 4.1 EXPERIMENT SETTINGS

**Datasets.** For evaluation, we employ two synthetic and one real-world dataset. The synthetic datasets (Greff et al., 2022) consist of numerous moving objects placed against complex backgrounds. MOVi-C contains up to 11 objects, whereas MOVi-E extends this to 23 objects and additionally incorporates linear camera motion. For real-world dataset, we adopt the YouTube-VIS (YTVIS) 2021 (Yang et al., 2021), which provides a diverse collection of video scenes sourced from YouTube.

**Evaluation Metrics.** We evaluate object discovery using Foreground Adjusted Rand Index (FG-ARI) and mean Best Overlap (mBO) (Seitzer et al., 2022). FG-ARI measures how well predicted masks align with ground-truth objects, excluding background pixels, and reflects segmentation quality and temporal consistency when computed over full videos. mBO, based on intersection-over-union (IoU), matches each ground-truth mask with the best corresponding prediction and averages the IoU, thereby assessing how accurately masks fit object boundaries. For both metrics, we report video-level scores (capturing consistency across time) as well as image-level scores (computed per frame).

Table 2: Experimental results on object dynamics prediction. Predictions are obtained by integrating each frozen pretrained model into the SlotFormer (SF) framework.

| Method | MOVi-C | | MOVi-E | | YouTube-VIS | |
|---|---|---|---|---|---|---|
| | FG-ARI↑ | mBO↑ | FG-ARI↑ | mBO↑ | FG-ARI↑ | mBO↑ |
| Reconstruction + SF | 50.7 | 25.9 | **70.6** | 24.3 | 27.4 | 28.9 |
| SlotContrast + SF | 63.8 | 26.1 | 70.5 | **24.9** | 29.2 | 29.6 |
| SRL (Ours) + SF | **68.9** | **27.4** | 70.4 | **24.9** | **32.2** | **30.0** |

**Implementation Details.** Following SlotContrast (Manasyan et al., 2025), we employ DINO-v2 (Oquab et al., 2024) as our backbone, using DINO-v2-Small/14 for MOVi-C and DINO-v2-Base/14 for MOVi-E and YTVIS. We resize input frames to $336\times336$ for MOVi datasets and $518\times518$ for YTVIS. We set the number of positive samples per anchor in denoising contrastive learning ($K$) to $8T$ for MOVi-C, $24T$ for MOVi-E, and $16T$ for YTVIS, respectively. The number of slots is set to 11, 15, and 7 for each dataset, following SlotContrast (Manasyan et al., 2025). The number of penalized slots ($M$) was consistently set to half the total slot count ($M = \lfloor S/2 \rfloor$), and loss coefficients $\lambda^{\text{CL}}$ and $\lambda^{\text{reg}}$ are set to 0.1 for all datasets.

## 4.2 COMPARISON TO STATE-OF-THE-ART METHODS

In Tab. 1, we evaluate SRL against state-of-the-art methods for video object-centric learning and observe consistent gains. As observed, our SRL improves reproduced SlotContrast[†] (Manasyan et al., 2025) by 5.5% (FG-ARI) and 8.8% (mBO) on MOVi-C. In addition, on the real-world YTVIS dataset, our method is even more effective, enhancing SlotContrast[†] by 18.5% (FG-ARI) and 8.2% (mBO). These results validate that SRL enhances FG-ARI by (1) promoting clear semantic boundaries and (2) encouraging one-to-one slot-object assignments, while mBO is also greatly improved by deblurring the object boundaries. On synthetic datasets, despite our superior performance on FG-ARI, our model achieves a comparatively lower mBO score than VideoSAUR (Zadaianchuk et al., 2023). We attribute this to VideoSAUR's specialized, motion-centric training objective, which is well-aligned with the primary characteristics of these datasets. The objects in the MOVi datasets exhibit highly constrained degrees of freedom; they are non-deformable, and their motion is restricted to rigid transformations such as translation and rotation. Thus, VideoSAUR's learning process is tailored to such monotonous scenario, which excels at grouping patches with consistent motion. This directly translates to more precise boundary segmentation and, consequently, a higher mBO score in this controlled environment. However, our model's strong performance on the more challenging YTVIS, which features objects with higher degrees of freedom and non-rigid deformations, demonstrates the effectiveness of SRL in capturing diverse and complex object boundaries, suggesting greater generalizability to real-world scenarios. Qualitative results are in the Appendix.

## 4.3 OBJECT DYNAMICS PREDICTION

To test whether our method benefits downstream tasks, we evaluate our pretrained video object-centric models on an object dynamics prediction task. Following SlotContrast, we attach a dynamic module on top of the frozen object-centric encoder and train it to predict future slots. We adopt SlotFormer (Wu et al., 2022) for the dynamic module, which performs multiple rollout steps to infer the dynamics of object slots after a set of burn-in frames.

For a fair comparison, we use the identical experimental setup introduced in SlotContrast. Experiments are conducted on MOVi-C, MOVi-E, and YTVIS-2021, and results are reported in Tab. 2. Across all datasets, our method consistently outperforms both the reconstruction-only baseline and SlotContrast, demonstrating that SRL yields object-centric features more amenable to modeling temporal evolution. Note that the performance on MOVi-E is nearly saturated, so the results do not differ significantly across methods. These results suggest that SRL not only improves static object discovery but also produces representations that better capture object dynamics in realistic video settings.

## 4.4 ABLATION STUDY

All studies are conducted on the MOVi-C dataset.

**Component Ablation** In Tab. 3, we examine the impact of each component, using our re-implemented SlotContrast as the baseline. Introducing the decoder deblurring objective ($\mathcal{L}^{\text{CL-dec}}$) provides a substantial boost in mBO, increasing it to 33.2. This result validates the objective's mechanism: by explicitly penalizing ambiguity at object boundaries, it compels the decoder to produce sharper, more precise segmentation masks. This enhanced boundary accuracy leads to a higher IoU with the ground truth, which is the basis of the mBO metric. Conversely, activating the encoder denoising objective ($\mathcal{L}^{\text{CL-enc}}$) yields a notable improvement in FG-ARI. By aligning the noisy patches correctly, the model achieves a more coherent and temporally stable clustering of foreground pixels.

Crucially, the full synergistic potential of our SRL is unlocked when they are built upon the foundation laid by our slot regularization. This initial regularization establishes a well-differentiated semantic space by minimizing the overlap between slot representations. By ensuring that each slot is initialized with a distinct object-level concept, we prevent the subsequent denoising and deblurring objectives from operating on fragmented representations where the model would inadvertently learn to sharpen the semantic boundaries between object fragments.

Table 3: Component ablation study.

| Deblur $\mathcal{L}^{\text{CL-dec}}$ | Denoise $\mathcal{L}^{\text{CL-enc}}$ | Reg $\mathcal{L}^{\text{reg}}$ | FG-ARI↑ | mBO↑ |
|:---:|:---:|:---:|:---:|:---:|
| | | | 70.8 | 31.4 |
| ✓ | | | 70.0 | 33.2 |
| | ✓ | | 72.2 | 31.2 |
| ✓ | ✓ | | 70.7 | 35.1 |
| ✓ | | ✓ | 73.0 | 33.5 |
| | ✓ | ✓ | 74.2 | 33.2 |
| ✓ | ✓ | ✓ | 74.3 | 34.5 |

**Effectiveness of Hierarchy in Contrastive Learning** To validate the necessity of our hierarchical design, we compare its results against two simplified, single-level variants in Tab. 4; the second row uses only the primary positive set ($\mathcal{P}$) and treats all other patches, including semi-positives, as negatives, while the third row uses only the semi-positive set ($\mathcal{Q}$) as the sole source of positive signal.

Both simplifications lead to a degradation in performance, but for different reasons. The positive-only variant suffers from a severe false negative problem; it incorrectly penalizes patches that belong to the same object but are not among the highest-confidence anchors (e.g., not Top-$K$ similar for the encoder, nor the anchor itself for the decoder). This corrupts the semantic space and leads to fragmented representations. The semi-positive-only variant is also suboptimal, as it forces one module to exclusively mimic the other's potentially flawed representation without a stable grounding signal. For instance, it would compel the decoder to perfectly replicate the encoder's

Table 4: Ablation study of hierarchical contrastive objective. Pos., S.Pos., and Time indicate whether the positive set $\mathcal{P}$, the semi-positive set $\mathcal{Q}$, and the temporal sampling strategy are used or not.

| Pos. | S.Pos. | Time | FG-ARI↑ | mBO↑ |
|:---:|:---:|:---:|:---:|:---:|
| ✓ | ✓ | ✓ | 74.3 | 34.5 |
| ✓ | | ✓ | 67.2 | 32.4 |
| | ✓ | ✓ | 69.9 | 32.7 |
| ✓ | ✓ | | 72.0 | 34.4 |

sharp but noisy groupings, preventing it from learning a more spatially coherent mask. These results confirm the necessity of our hierarchical structure.

**Importance of Temporal Context in Contrastive Sampling** Our framework gathers positive and semi-positive candidates from all $T$ frames available in a video clip. To investigate the importance of this temporal context, we conduct an ablation where contrastive sets are sourced exclusively from an anchor's current frame. The results in Tab. 4 (last row) reveal that the impact on mBO is marginal since the blurring effect is primarily the spatial phenomenon, yet the semantic clustering (FG-ARI) benefits immensely from temporal context. Therefore, to achieve robust and temporally-consistent predictions in videos, we claim that it is crucial to leverage a temporal window.

**Number of positive patches for denoising contrastive learning $K$** In Fig. 3a, We study the sensitivity of our denoising contrastive learning module to the number of positive neighbors, $K$, used in the positive set $\mathcal{P}^{\text{enc}}$. On MOVi-C, the best results occur around $K = 8T$, striking a balance between semantic coverage and noise. Importantly, SRL is robust to the choice of $K$: performance fluctuates only marginally and consistently outperforms SlotContrast across a wide range of $K$.

**Number of penalized slots $M$** We also analyze the impact of $M$, the number of redundant slots penalized by our slot regularization during the warm-up phase, in Fig. 3b. This parameter determines how aggressively the model prunes overlapping slot assignments. Our analysis reveals that a simple yet effective heuristic, setting $M$ to half the total number of slots ($\lfloor S/2 \rfloor$), consistently achieves decent performances robustly preventing slot collapse. Thus, $M$ is set to $\lfloor S/2 \rfloor$ across all datasets.

**Loss coefficients** We analyze the sensitivity of the loss coefficients $\lambda^{\text{reg}}$ and $\lambda^{\text{CL}}$, in Fig. 3c and Fig. 3d. Results are reasonably stable near the default, but extreme values degrade performance.

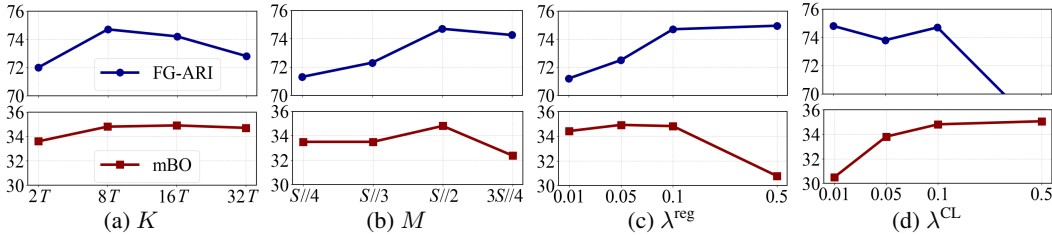

Figure 3: Ablation study on coefficients. For (c), we vary $\lambda^{\text{reg}}$ with fixed $\lambda^{\text{CL}}$, and vice versa for (d).

Increasing $\lambda^{\text{reg}}$ promotes slot uniformity (discouraging semantic overlap) yet acts as a smoothing prior that can blur boundaries and lower mBO; decreasing it too much under-constrains slots, inducing over-fragmentation and reducing FG-ARI. Increasing $\lambda^{\text{CL}}$ strengthens local discrimination and edge sharpening, but when set too high, it over-separates fine-grained features, lowering FG-ARI. On the other hand, making it too small lets reconstruction dominate (a low-pass effect), weakening edge cues and lowering mBO. Still, the hyperparameter choice is straightforward: we use a single fixed setting $\lambda^{\text{reg}} = \lambda^{\text{CL}} = 0.1$, which performs reliably across all datasets.

## 4.5 EFFECTIVENESS OF SYNERGISTIC LEARNING BETWEEN ATTN AND MASK

We qualitatively compare two distinct spatial maps, the slot attention maps **Attn** and decoder predictions **Mask**, on the MOVi-C dataset between SlotContrast (Manasyan et al., 2025) and our method in Fig. 4. As observed, SlotContrast frequently yields noisy **Attn** maps, which, when coupled with the decoder's blurry **Mask**, deteriorate a vicious cycle and lead to inconsistent and noisy slot representations. In contrast, our approach extends SlotContrast (Manasyan et al., 2025) by introducing synergistic objectives that facilitate mutual refinement between the **Attn**

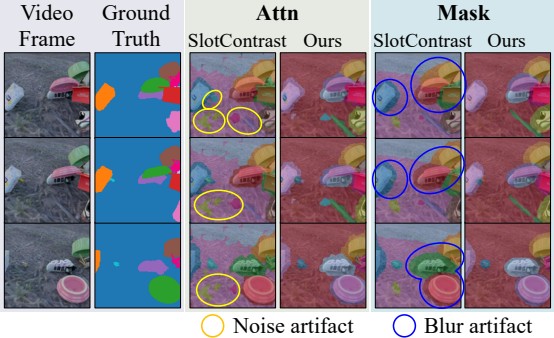

Figure 4: Visualization of **Attn** and **Mask**.

and **Mask** representations. This process leverages their complementary strengths, producing denoised and deblurred predictions. As a result, the two spatial maps become more consistent with one another, demonstrating the effectiveness of our synergistic learning framework.

## 4.6 SLOT SPECIALIZATION

To discourage multiple slots from redundantly capturing the same object representation, we introduce slot regularization during the warm-up stage of training. We assess its impact by visualizing predicted masks on the MOVi-C dataset, comparing the baseline with and without $\mathcal{L}^{\text{reg}}$ (Fig. 5). The visualization demonstrates that slot regularization reduces object over-fragmentation by encouraging greater disparity among slots that would otherwise collapse onto the same semantics. This promotes a more effective one-to-one correspondence between slots and objects, thereby strengthening the synergy of our overall representation learning framework.

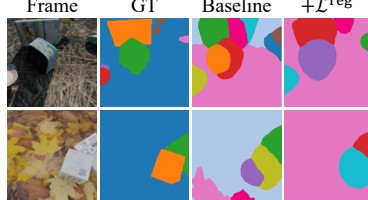

Figure 5: Visualization of decoder's final prediction, **Mask**.

## 5 CONCLUSION

We presented a novel framework that addresses a critical, previously overlooked bottleneck in video object-centric learning: the representational conflict between the encoder's sharp but noisy groupings and the decoder's coherent but blurry reconstructions. Our solution, Synergistic Representation Learning, introduces two purpose-built, ternary contrastive objectives that allow the encoder and decoder to enter a virtuous cycle of mutual refinement. The effectiveness of our approach, validated by state-of-the-art performance, demonstrates that explicitly modeling and resolving the discrepancies between a model's internal representations is a powerful mechanism for enhancing performance.

## ACKNOWLEDGEMENTS

This work was supported in part by MSIT/IITP (No. RS-2022-II220680, RS-2020-II201821, RS-2019-II190421, RS-2024-00459618, RS-2024-00360227, RS-2024-00437633, RS-2024-00437102, RS-2025-25442569), MSIT/NRF (No. RS-2024-00357729), and KNPA/KIPoT (No. RS-2025-25393280).

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

## A  VISUALIZATION

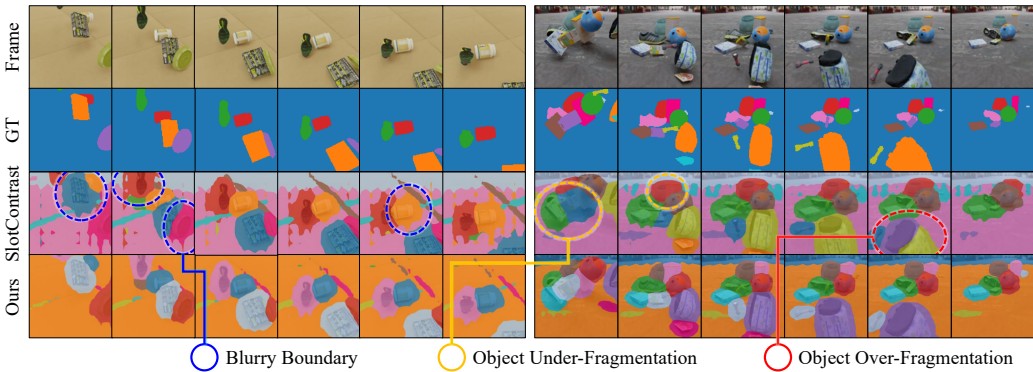

Figure A1: Qualitative comparison results of ours and SlotContrast on MOVi-C dataset.

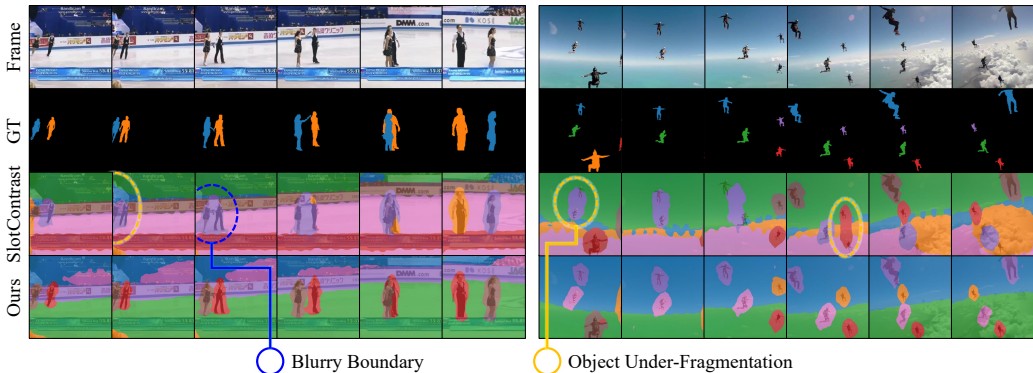

Figure A2: Qualitative comparison results of ours and SlotContrast on YTVIS 2021 dataset.

### A.1  QUALITATIVE RESULTS

For qualitative evaluation, we compare our method with SlotContrast (Manasyan et al., 2025) on the MOVi-C and YTVIS 2021 datasets, as shown in Fig. A1 and Fig. A2.

On the MOVi-C dataset, our method demonstrates a notable improvement in object separation. As illustrated in the left video example, our baseline (SlotContrast) often produces blurry decoded masks where slots exhibit diffuse spatial support, extending beyond the object's true boundaries. In contrast, our method generates compact slots that adhere more faithfully to the object's contours, resulting in clearer semantic boundaries. Furthermore, the right example shows how these sharp boundaries directly mitigate a common failure mode of object under-fragmentation (the erroneous grouping of multiple objects into a single slot). Whereas SlotContrast incorrectly merges distinct objects (e.g., regions covered by red and green slots), our SRL framework alleviates the under-fragmentation issue by partitioning them into different slots. Complementing this, our warm-up strategy prevents the opposing failure mode of over-fragmentation, where a single object is fragmented into different parts. Together, these components ensure a more robust one-to-one correspondence between slots and objects.

This trend extends to the more challenging YTVIS dataset (Fig. A2). The baseline's inability to maintain compact semantic boundaries causes it to fail in scenarios with object overlap, where proximal entities are often merged into a single slot. For instance, in both video examples, SlotContrast incorrectly assigns one slot to cover two distinct people (the region covered by the purple slots in both videos). In contrast, our method yields sharper predictions by learning to clarify the semantic boundary via denoising and deblurring contrastive objectives. This allows slots to more faithfully specialize to individual objects, even when they overlap.

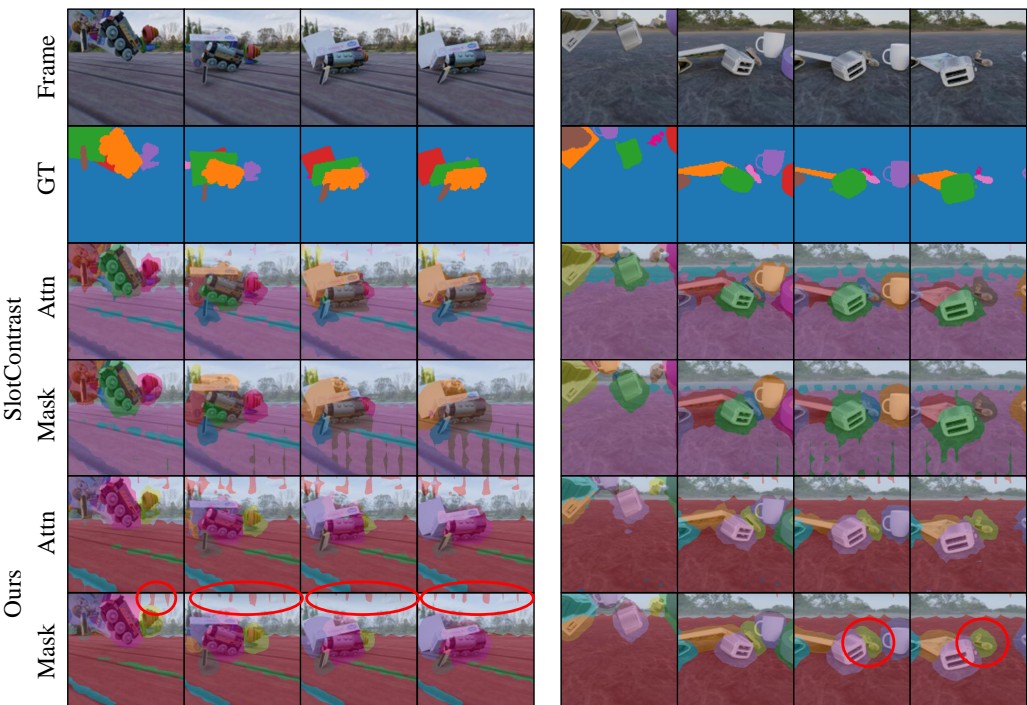

Figure A3: Failure cases on MOVi-C dataset.

## A.2 Failure Cases

To provide a more complete analysis of our method's behavior and limitations, we additionally visualize representative failure cases in Fig. A3. Specifically, we present two samples from the MOVi-C dataset to qualitatively examine the scenarios where our model performs suboptimally.

In this work, we focused on the discrepancy that encoder spatial maps tend to be noisy while decoder maps are overly blurry. However, as shown in Fig. A3 (left), there exist cases where the noise originating from the encoder propagates into the decoder and persists even after training (red circles). This occurs because SRL is primarily designed to address the dominant issue of blurry decoded masks, and does not explicitly regularize decoder-side noise. As a result, certain noisy attention patterns may remain, similarly to the SlotContrast baseline. Although our method still alleviates over-fragmentation and reduces blur in such cases, explicitly modeling and suppressing this propagated noise remains an important direction for future work.

In addition, in Fig. A3 (right), we observe that our method occasionally under-fragments extremely small objects, failing to allocate a dedicated slot to each of them. This indicates that our model remains vulnerable when the targets are very small. We believe that explicitly targeting small-object discovery and representation is another promising direction for future work.

## B Further Experiments

### B.1 MAE Loss for Reconstruction

In the main manuscript, our SRL framework is designed to address the inherent weakness of the commonly used MSE reconstruction loss, namely, its tendency to produce blurred outputs, which causes a vicious cycle during training. To examine whether similar vulnerabilities arise under alternative reconstruction objectives, and to assess the robustness of SRL beyond the MSE setting, we additionally replace MSE with

Table B1: Experimental results using MAE loss for reconstruction.

| Method | FG-ARI↑ | mBO↑ |
|---|---|---|
| SlotContrast | 73.24 | 27.54 |
| SRL (Ours) | 74.57 | 34.28 |

MAE loss and evaluate both SlotContrast and our SRL on MOVi-C dataset. The quantitative and qualitative results are summarized in Tab. B1 and Fig. B1, respectively.

Compared to using MSE as the reconstruction loss, MAE tends to emphasize the majority of pixels, which makes it robust to certain high error patches. While this alleviates the over-fragmentation and improves FG-ARI, this majority-focused behavior often causes under-fragmentation of small objects and amplifies irregular noise patterns, as shown in Fig. B1, which in turn degrades mBO. Nonetheless, since our method excels at denoising such noise patterns, it is shown that SRL consistently improves the performances. In particular, the substantial gain in mBO indicates that our approach reduces noisy activations even when the reconstruction objective is modified from MSE to MAE. These results demonstrate that the denoising and deblurring benefits of SRL are applicable to various challenging scenarios across different objectives (*i.e.,* MSE or MAE).

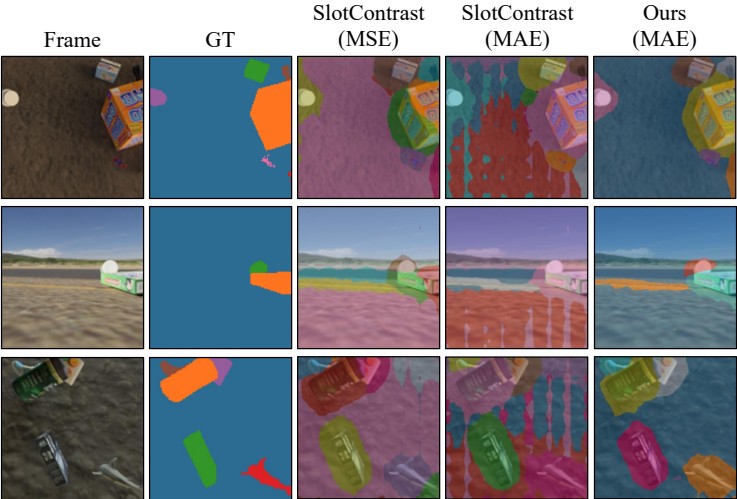

Figure B1: Qualitative comparison results when utilizing MAE loss for reconstruction objective.

## B.2 EXPERIMENTS ON ADDITIONAL DATASETS

To assess whether SRL generalizes beyond the datasets used in the main manuscript, we evaluate on the DAVIS 2017 (Pont-Tuset et al., 2017) validation set ($37 \times 37$ patch grid) using a model trained on YTVIS-2021, following the transfer protocol introduced in VideoSAUR (Zadaianchuk et al., 2023). We report the boundary F-score $\mathcal{F}$ and Jaccard index $\mathcal{J}$ in Tab. B2. SRL achieves substantial improvements over SlotContrast, improving $\mathcal{J}$ by +11.7 points and the combined score $\mathcal{J}\&\mathcal{F}$ by +7.5 points.

We further evaluate on YouTube-VIS 2019 (YTVIS-2019) under two scenarios: (1) cross-dataset transfer from the model trained on YTVIS-2021, and (2) in-dataset evaluation, where the model is trained on the YTVIS-2019 train set and evaluated on its validation set. As summarized in Tab. B3 and Tab. B4, SRL consistently surpasses SlotContrast across both settings, yielding improvements on ARI and mBO regardless of whether the model is transferred or trained in-domain.

Taken together, these results demonstrate that SRL generalizes robustly across video domains and dataset shifts, providing stronger object-centric representations than SlotContrast in both transfer and in-distribution evaluations. This suggests that the proposed learning signal not only enhances grouping quality within the training domain but also yields transferable object discovery behavior that extends to diverse video benchmarks.

## B.3 EXPERIMENTS ON DIFFERENT PRETRAINED BACKBONES

To assess whether SRL remains effective when applied to backbone encoders beyond DINO-v2, we replace DINO-v2 with either Franca (Venkataramanan et al., 2025) or MoSiC (Salehi et al.,

Table B2: Experimental results on DAVIS dataset.

| Method | $\mathcal{F}$ | $\mathcal{J}$ | $\mathcal{F}\&\mathcal{J}$ |
|---|---|---|---|
| SlotContrast | 22.2 | 36.5 | 29.3 |
| SRL (Ours) | 25.4 | 48.2 | 36.8 |

Table B3: YTVIS2019 results trained on YTVIS2021 dataset.

| Method | FG-ARI↑ | mBO↑ |
|---|---|---|
| SlotContrast | 16.6 | 43.3 |
| SRL (Ours) | 20.4 | 53.3 |

Table B4: YTVIS2019 results trained on YTVIS2019 dataset.

| Method | FG-ARI↑ | mBO↑ |
|---|---|---|
| SlotContrast | 16.7 | 44.9 |
| SRL (Ours) | 19.1 | 46.9 |

Figure B2: Qualitative comparison when using MoSiC as backbone encoder. Blue circles indicate position bias.

Table B5: Experiments on MOVi-C dataset on Franca ViT-B/14.

| Method | FG-ARI↑ | mBO↑ |
|---|---|---|
| SlotContrast | 66.8 | 35.6 |
| SRL (Ours) | 66.1 | 37.2 |

Table B6: Experiments on YTVIS-2021 dataset on Franca ViT-B/14.

| Method | FG-ARI↑ | mBO↑ |
|---|---|---|
| SlotContrast | 35.3 | 32.7 |
| SRL (Ours) | 38.9 | 36.4 |

Table B7: Experiments on MOVi-C dataset on MoSiC ViT-B/14.

| Method | FG-ARI↑ | mBO↑ |
|---|---|---|
| SlotContrast | 70.3 | 31.6 |
| SRL (Ours) | 74.3 | 37.3 |

2024), and evaluate both SlotContrast and SRL under the same training and evaluation protocol. The experimental results on the MOVi-C dataset are reported in Tab. B5- B6 and Tab. B7, respectively.

Across both backbones, SRL consistently improves performance. When using Franca as the backbone, SRL is particularly beneficial on the mBO metric on MOVi-C, and surpasses SlotContrast by a large margin on YTVIS 2021. SRL is also effective when built on MoSiC, a denoised backbone specifically designed to reduce feature-level noise. Interestingly, we observe that such denoised backbones may introduce new artifacts: MoSiC features often exhibit a strong positional bias, where slots collapse onto empty background regions or fail to track moving objects. This suggests that part of the denoising effect comes at the cost of distorted spatial structure (see Fig. B2). Nonetheless, our method effectively mitigates these noisy artifacts and restores meaningful object assignments, thereby achieving a large performance uplift over SlotContrast. These results confirm that SRL not only transfers across datasets but also remains robust across diverse backbone architectures, even those subject to substantial feature-level modifications.

## B.4 EXPERIMENTS ON IMAGE DATA

Our SRL is applicable to static images, as the conflict between the encoder's sharpness and decoder's smoothness exists in the slot attention architecture itself, independent of temporal dimensions. Therefore, we evaluate SRL on the MSCOCO 2017 dataset (Lin et al., 2014) using the same training protocol as the baseline. The

Table B8: Results on COCO.

| Method | ARI↑ | mBO↑ |
|---|---|---|
| Baseline | 40.5 | 28.8 |
| Ours | 42.8 | 29.4 |

experiments are conducted with DINO-v2-Small/14. As shown in Tab. B8, SRL achieves an improvement of +2.3 in ARI and +0.6 in mBO over the reconstruction-only baseline. This demonstrates that even without temporal cues, the mutual refinement between the encoder's sharp attention and the decoder's spatial coherence effectively improves object discovery. While our study focused on video benchmarks, these additional results confirm that SRL is a generalizable solution for object-centric learning.

(a) Ablation study for varying the end iteration of $\mathcal{L}^{\text{reg}}$ (b) Ablation study on varying the start iteration of $\mathcal{L}^{\text{CL}}$

Figure B3: Ablation study on the staged training boundaries. (a) Varying the iteration at which slot regularization is turned off. (b) Varying the iteration at which contrastive learning objectives are activated.

## B.5 ADDITIONAL ABLATION STUDY

We conduct ablation studies to examine the robustness of our staged training strategy, which consists of: (i) an early slot regularization phase, and (ii) a later contrastive learning phase (denoising/deblurring). The model is trained for 100k iterations on the MOVi-C dataset, and we vary the transition point of each stage while keeping the remainder of the training settings identical. The results are summarized in Fig. B3.

**When to Stop Slot Regularization.** We first vary the iteration at which slot regularization is disabled, shown in Fig. B3 (a). For reference, the SlotContrast baseline achieves 70.8 for FG-ARI and 31.4 for mBO on this benchmark. Across all tested schedules, our method substantially exceeds the baseline, and the resulting performance curves remain smooth after 10k iterations. Stopping the regularization slightly later (e.g., around 20k iterations) yields a modest improvement in mBO, indicating that the method does not rely on a finely tuned early cutoff. Overall, SRL maintains strong FG-ARI and mBO performance across different regularization stopping points.

**When to Start Synergistic Representation Learning.** Next, we vary when the contrastive learning objectives are activated during training. The results are illustrated in Fig. B3 (b). Once again, all tested configurations clearly outperform SlotContrast by a significant margin. We attribute this to the need for the encoder and decoder to first learn reasonably stable spatial representations; if contrastive learning is applied too early, the two branches end up guiding each other based on poorly formed features, whereas after roughly 20k iterations, the representations have largely converged and provide reliable signals. Nonetheless, a relatively broad starting interval still yields competitive results, indicating that the contrastive learning module is not overly sensitive to the precise activation point.

These ablations demonstrate that SRL is robust to the choice of stage boundaries. The method consistently improves over SlotContrast under all tested schedules, and performance behaves smoothly rather than collapsing when deviating from an optimal configuration. This indicates that SRL offers a stable and reliable training procedure that does not require careful tuning of the transition point.

## B.6 VICIOUS CYCLE

To investigate whether the vicious cycle between attention noise and mask blur indeed arises during training, we conduct a qualitative analysis on the MOVi-C dataset by visualizing both attention maps and masks at the early and converged stages of training. The visualizations are provided in Fig. B4.

For SlotContrast, we observe that the quality of both attention and masks can deteriorate as training progresses. In the left example of Fig. B4, objects that are initially well separated gradually lose their semantic boundaries, causing multiple objects to be merged into a single slot (red circle). In the right example, blurred boundaries prevent the model from disentangling overlapping objects, and residual attention noise persists even after training, propagating into the decoder masks.

In contrast, our method effectively suppresses this error propagation. In the left example, even when some objects are under-segmented at early stages, the deblurring process of semantic boundaries encourages the model to recover clear object-wise separation as training proceeds. In the right example, although the encoder attention maps initially exhibit noisy and blurred boundaries, our method progressively removes this noise and yields sharper encoder attention and cleaner decoder masks by the end of training.

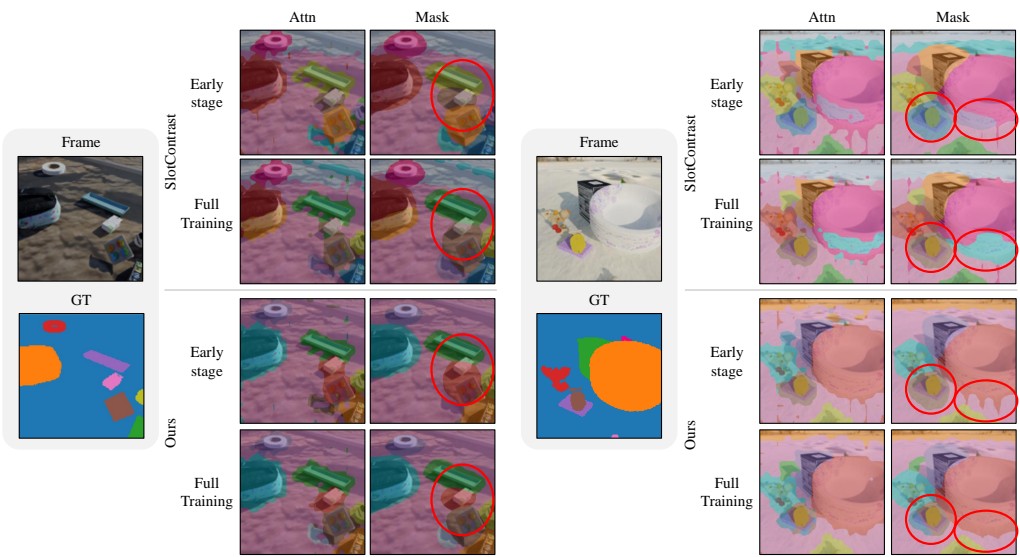

Figure B4: Qualitative analysis for vicious cycle.

## C  BROADER IMPACTS

The advancements presented in this work have significant potential for positive societal impact by enhancing the capabilities of machines to understand and interact with the dynamic world in a more human-like, object-centric manner. By enabling robust unsupervised object discovery and tracking, our SRL can power more effective, efficient, and accessible tools for a wide range of applications without requiring costly human annotations.

However, the improved capabilities for unsupervised object tracking and segmentation could be repurposed for malicious uses. A primary concern is the potential for enhanced surveillance and monitoring. A system that can reliably identify and track distinct objects without supervision could be deployed in mass surveillance systems without the subject's consent, raising significant privacy concerns.

In addition, the synergistic refinement, which is the core principle of our work, suggests a generalizable paradigm for other foundational architectures beyond object-centric learning, where the framework consists of encoder-decoder architectures. For instance, the training dynamics of Generative Adversarial Networks (GANs) exhibit a similar discrepancy between the representations of the discriminator and the generator. While the generator's features are semantically coherent enough to produce segmentation masks (Zhang et al., 2021), the discriminator has been observed to lose the semantic information as training progresses (Chen et al., 2019). Yet, the discriminator learning useful semantics has proven beneficial for stable GAN training (Chen et al., 2019). Therefore, we posit that the feature discrepancy in encoder-decoder architectures can be leveraged as a complementary training signal in other domains as well.

## D  THE USE OF LARGE LANGUAGE MODELS (LLMS)

We used an LLM-based writing assistant solely for language refinement, including grammar correction, phrasing improvements, and ensuring clarity. The model did not generate ideas, analyses, experiments, or results. All technical content was authored and verified by the authors, who take full responsibility for the manuscript. We affirm that no proprietary data beyond the text itself was shared with the writing tool.

