# OpenReview forum: "From Vicious to Virtuous Cycles: Synergistic Representation Learning for Unsupervised Video Object-Centric Learning"
_ICLR.cc/2026/Conference — ICLR 2026 Poster_

### Official Review · Reviewer_n3E7 · 2025-10-26

**Soundness:** 2
**Presentation:** 3
**Contribution:** 2
**Rating:** 4
**Confidence:** 4

**Summary:**

The paper introduces three novel approaches to improving the quality of segmentation maps in Slot-Attention based approaches. The authors note that the attention maps from the Slot Attention mechanism are sharp, but noisy, while the decoder provides maps that are not noisy but overly smooth. To alleviate these issues the authors propose adding two hierarchical contrastive losses over constructed sets of positive, semi-positive and negative feature map patches constructed from the segmentation maps from the Slot Attention and Decoder modules. The encoder’s contrastive loss tries to align the feature map prior to slot attention with the feature map from the decoder. The decoder similarly tries to align with patches from the encoder. Additionally, the authors propose an approach to alleviate over-clustering by regularizing slots. Overall, these approaches lead to gains on the reported segmentation metrics.

**Strengths:**

The paper describes the proposed approaches in a clear manner and performs sufficient ablations to show that each component contributes to improving segmentation performance. The presentation is good and the figures help to understand the method.

**Weaknesses:**

The paper should be more specific in their experimental setup.

It is unclear if the poor decoder segmentation maps are a result of the choice of MLP, or if other types of decoders suffer the same issue.

Furthermore, it is unclear if this is a result of MSE, which the paper implies, and if it can be resolved through the use of a different reconstruction loss.

It should be stated which version of DINOv2 was used.

The authors make claims of the gradients flowing from the decoder being “corrupted” and
“low-frequency”. It would be good if the authors could give some evidence to back up this statement.

The paper does not provide examples of cases where the proposed approach fails and leaves room for improvement.

Some of the writing is superfluous. For example: “This is because under MSE, the loss is calculated from the square of the error, meaning that the penalty grows quadratically with the error’s magnitude.”

**Questions:**

- Do you use an MLP decoder as is alluded to in the introduction?
- Do other types of decoders suffer from the same overly smooth segmentation maps that you try to alleviate?
- Does using a different loss, e.g. L1, for the reconstruction objective, rather than MSE, lead to less blurry decoder segmentations?

---

> ### Author Response · Authors · 2025-11-29
>
> ### **W1: The paper should be more specific in their experimental setup. It should be stated which version of DINOv2 was used.**
>
> We sincerely thank the reviewer for detailed feedback. We have updated our manuscript to indicate the type of backbones for each dataset.
>
> ### **W2: Do you use an MLP decoder as is alluded to in the introduction? It is unclear if the poor decoder segmentation maps are a result of the choice of MLP, or if other types of decoders suffer the same issue.**
>
> Yes, we use the standard MLP-based slot decoder used in Slot Attention and SlotContrast, in order to keep our setup consistent with prior work and to isolate the effect of SRL itself.
> Generally, the transformer decoder is known for its tendency to produce tighter masks, but it suffers from different challenges [A].
> Specifically, transformer decoders become less suitable as the feature resolution increases and suffer from overusing slots to split a single object into multiple parts. Also, transformer decoders require more careful tuning for each dataset.
> This matches our own observations: swapping the MLP for the transformer decoder can slightly sharpen boundaries, but it suffers from splitting a single object into multiple parts (higher mBO but lower FG-ARI).
>
> **Experimental results with Transformer decoder**
> >| Method | FG-ARI | mBO |
> >| --- | :---: | :---: |
> >| SlotContrast | 62.6 | 41.2 |
> >| SRL (Ours)  | 65.0 | 40.1 |
>
> In such cases, the effect of our deblurring contrastive learning on sharpening boundaries can be diminished, despite the overall performance gains being consistent.
> Nevertheless, recent slot-based works report that MLP decoders are generally much easier to adapt across datasets and scale more gracefully to higher-resolution visual inputs than more complex decoder architectures, which demonstrates that using the MLP decoder is the mainstream [A].
> This makes it practically important to address the characteristic blurriness of MLP decoders rather than avoid them, and our method is designed precisely to play this role by explicitly deblurring decoder maps using the sharper encoder signals.
>
> [A] Bridging the Gap to Real-World Object-Centric Learning. ICLR 2023
>
> ### **W3: Furthermore, it is unclear if this is a result of MSE, which the paper implies, and if it can be resolved through the use of a different reconstruction loss. (Does using a different loss, e.g., L1, for the reconstruction objective, rather than MSE, lead to less blurry decoder segmentations?)**
>
> We appreciate the reviewer’s question about the role of the reconstruction loss.
>
> In the paper, we argue that standard MSE encourages the decoder to favor safe but blurry solutions: large deviations are quadratically penalized, so averaging over ambiguous regions is often the easiest way to reduce the loss.
>
> To check whether this phenomenon can simply be removed by changing the reconstruction loss, we replaced MSE with an L1/MAE loss and re-ran the experiments.
>
> **Experimental results with MAE loss as reconstruction objective**
> >| Method | FG-ARI | mBO |
> >| --- | :---: | :---: |
> >| SlotContrast | 73.2 | 27.5 |
> >| SRL (Ours) | 74.6 | 34.3 |
>
> Compared to using MSE as the reconstruction loss, MAE tends to emphasize the majority of pixels, which makes it robust to certain high error patches.
> While this alleviates the over-fragmentation and improves FG-ARI, this majority-focused behavior often causes under-fragmentation of small objects and amplifies irregular noise patterns, as shown in Fig. B1 (Appendix), which in turn degrades mBO.
>
> Still, we observe that SRL remains effective under MAE: it consistently improves both FG-ARI (+1.3) and mBO (+6.7) over MAE+SlotContrast, suppressing the additional noise and yielding cleaner object separation. In other words, the encoder–decoder discrepancy we target is not specific to the MSE objective; it persists under MAE, and SRL provides complementary gains for both reconstruction losses.

---

> ### Author Response · Authors · 2025-11-29
>
> ### **W4: The authors make claims of the gradients flowing from the decoder being “corrupted” and “low-frequency”. It would be good if the authors could give some evidence to back up this statement.**
>
> We thank the reviewer for pointing out the ambiguity in our wording, and we apologize for the confusion. Our intention was not to claim that the gradients themselves are corrupted but rather that, under an MSE reconstruction objective computed on already blurry decoder outputs, the optimization is biased toward recovering low-frequency content.
> This phenomenon occurs because sharp, high-frequency features are typically absent in the decoded representation for two primary reasons.
>
> First, due to the mean-seeking property of MSE [A], the decoder acts as a low-pass filter, failing to express high-frequency variations in the decoded output. This implies that the sensitivity of the decoder to high-frequency changes is negligible.
> In addition, according to [B], compressed features often lack the details regarding the precise location of all details, so that reconstructing from these compressed versions leads to blurry output.
>
> Consequently, since the decoder cannot reconstruct high-frequency details, the error signal derived from the reconstruction loss is inherently biased toward recovering low-frequency content.
>
> [A] Image-to-Image Translation with Conditional Adversarial Networks. CVPR 2017.
>
> [B] Generating Images with Perceptual Similarity Metrics based on Deep Networks. NeurIPS 2016.
>
> ### **W5: The paper does not provide examples of cases where the proposed approach fails and leaves room for improvement. (failure mode)**
>
> We have updated our Appendix with the common failure modes.
>
> In this work, we focused on the discrepancy that encoder spatial maps tend to be noisy while decoder maps are overly blurry.
> We acknowledge, however, that there are cases where the noise originating from the encoder can propagate into the decoder and persist even after training.
> This occurs because SRL is primarily designed to address the dominant issue of overly smooth, blurry decoded masks and does not explicitly regularize decoder-side noise.
> As a result, certain noisy attention patterns may remain, similarly to the SlotContrast baseline.
> Although our method still alleviates over-fragmentation and reduces blur in such situations, explicitly modeling and suppressing this propagated noise remains an important direction for future work.
> We also observe that our method can occasionally under-fragment extremely small objects, failing to allocate a dedicated slot to each of them.
> This indicates that our model remains vulnerable when the targets are very small.
> We believe that explicitly targeting small-object discovery and representation is another promising and complementary direction for future work.
>
> Qualitative analysis is updated in the revised version.
>
> ### **W6: Some of the writing is superfluous. For example: “This is because under MSE, the loss is calculated from the square of the error, meaning that the penalty grows quadratically with the error’s magnitude.”**
>
> We agree that some descriptions were superfluous for the ICLR audience. We have removed the superfluous explanations, including the details on MSE, to improve the paper's conciseness.

---

### Official Review · Reviewer_18N8 · 2025-10-27

**Soundness:** 2
**Presentation:** 2
**Contribution:** 3
**Rating:** 4
**Confidence:** 5

**Summary:**

The authors identified an objectively existing phenomenon, which I believe widely observed but never addressed by most OCL researchers (including me):
- The slot attention maps tend to segment the visual scene with noises;
- while the decoder attention maps tend to segment it with blurred boundaries.

Then they formulate the issues behind it:
- Noisy Encoder Reinforces the Decoder’s Blurriness;
- Blurry Decoder Corrupts the Encoder’s Learning Signal.

Accordingly, the authors propose two ternary contrastive losses that make the most of intrinsic supervision signals between the slot attention and decoding. Specifically,
- The positive pseudo labels, which are the overlap between the slot attention and decoding;
- The semi-positive pseudo labels, which are the inconsistent segmentations between the slot attention and decoding;
  - Differently chosen for their two losses
- The negative pseudo labels, which are the different segmentations between the slot attention and decoding.

Respectively, they designed two losses to achieve:
- DENOISING CONTRASTIVE LEARNING: REFINING THE ENCODER VIA DECODER COHERENCE
- DEBLURRING CONTRASTIVE LEARNING: REFINING THE DECODER VIA ENCODER SHARPNESS

Combined with their novel slot regularization, under staged training, their method achieved sota on video datasets in object discovery tasks.

**Strengths:**

Intuitve formulation on a widely observed phenomenon and feasible solutions.

**Weaknesses:**

(Writing issues first but not most important)

W1
---
Line 017 "We identify that this discrepancy gives rise to a vicious cycle; the noisy ...": ";" should be ":".

W2
---
Line 248 Equation (4) vs (6): Compared with (6), Equation (4) seems missing the $\Sigma$ and $\frac{1}{|P|}$ on the first term.

W3
---
Line 276, 287 Equation (6) and (7): Should be written in one Equation, not two.

W4
---
Line 144-145 "representational conflict between the slot attention maps and **reconstruction maps**": "reconstrcution maps" is a misleading term (First sight, I was wondering if you are mentioning the reconstructed feature maps?), better to use ones like "reconstraction/decoding attention maps".

W5
---
Sect 3.1, "Noisy Encoder Reinforces the Decoder’s Blurriness" and "Blurry Decoder Corrupts the Encoder’s Learning Signal": These two parts lack mathematical or statistical analysis/probing. Providing such analyses will surely make your problem definition more convincing.

W6
---
Sect Experiment. No OCL results on image datasets. Why not testing on image datasets like COCO? Theoretically your method supports OCL on both images and videos.

W7
---
More hyperparameters, Line 341, i.e., $\lambda reg$ and $\lambda CL$, and top-$K$, as well as $M$ at some where. Anyway, not a big deal. Most top conference fancy works would introduce one or two hyperparameters.

W8
---
More computation cost. According the two contrastive losses, pairwise cosine similarities need to be calculated and the three subsets need to be determined. So there must be most latency and VRAM consumption. It is necessary to provide quantitative results on these.

W9
---
Slot regularization contributes a large part to the total performance gain as shown in Tab 2. But this is not included in the main storyline.

W10
---
Slot regularization is similar to slot pruning-related techniques, which can boost the performance almost in any cases. Thus related works should be compared or at least discussed:
- SOLV: Self-supervised Object-Centric Learning for Videos. NeurIPS 2023
- MetaSlot: Break Through the Fixed Number of Slots in Object-Centric Learning. NeurIPS 2025.

W11
---
The proposed contrastive losses are actually a kind of **fine-grained** ***self-distillation***, compared with the following existing works. Thus related works should be compared or at least discussed:
- SPOT: Self-Training with Patch-Order Permutation for Object-Centric Learning with Autoregressive Transformers. CVPR 2024.
  - offline "self"-distillation on slot attention masks
- DIAS: Slot Attention with Re-Initialization and Self-Distillation. ACM MM 2025.
  - online self-distillation on slot attention masks
- SlotMatch: SlotMatch: Distilling Temporally Consistent Object-Centric Representations for Unsupervised Video Segmentation.
  - offline "self"-distillation on slots
- SmoothSA: Smoothing Slot Attention Iterations and Recurrences.
  - online self-distillation on slots

Interesting works are worth encouragement. So I will change my ratings if the authors can address my concerns.

**Questions:**

Please refer to the former part.

---

> ### Author Response · Authors · 2025-11-29
>
> ### **W1, W3, W4: Minor Suggestions**
> - Line 017 "We identify that this discrepancy gives rise to a vicious cycle; the noisy ...": ";" should be ":".
> - Line 276, 287 Equation (6) and (7): Should be written in one Equation, not two.
> - Line 144-145 "representational conflict between the slot attention maps and reconstruction maps": "reconstruction maps" is a misleading term (First sight, I was wondering if you are mentioning the reconstructed feature maps?), better to use ones like "reconstruction/decoding attention maps".
>
> We appreciate the reviewer for constructive feedback. These suggestions are updated in the manuscript. For the term “reconstruction maps”, we have used “decoded output masks after reconstruction” instead.
>
>
> ### **W2: Line 248 Equation (4) vs (6): Compared with (6), Equation (4) seems missing the $\Sigma$ and $\frac{1}{|P|}$ on the first term.**
>
> We thank the reviewer for carefully checking the equations. The first term in Eq. (4) is missing the $\Sigma$ and $\frac{1}{|\mathcal{P}|}$ by design.
> In Eq. (4), the first term corresponds to the case where the positive set consists of a single positive patch, namely the original encoder feature at position $(t,i)$.
> In other words, $|P_{t,i}^{\text{dec}}| = 1$ and $P_{t,i}^{\text{dec}}={(t,i)}$.
> In this case, the normalization factor $\frac{1}{|P_{t,i}^{\text{dec}}|}$ is simply $1$, and the summation over $P_{t,i}^{\text{dec}}$ collapses to a single term.
> For this reason, we write the first term in Eq. (4) directly in its single-positive form without an explicit ($\Sigma$) or ($\frac{1}{|P|}$).
> By contrast, Eq. (6) is written in the general multi-positive case, where $|P_{t,i}^{\text{enc}}| > 1$, so the summation and $\frac{1}{|P_{t,i}^{\text{enc}}|}$ factor appear explicitly. To avoid confusion, we will clarify in the revised manuscript that the first term in Eq. (4) corresponds to a single-positive instance of the more general form in Eq. (6).
>
> ### **W5: Sect 3.1, "Noisy Encoder Reinforces the Decoder’s Blurriness" and "Blurry Decoder Corrupts the Encoder’s Learning Signal": These two parts lack mathematical or statistical analysis/probing. Providing such analyses will surely make your problem definition more convincing.**
>
> We thank the reviewers for pointing out the need for clarification of the vicious cycle.
>
> To justify the claim that a noisy encoder reinforces blurriness, we rely on the fundamental statistical property of the MSE reconstruction objective, often referred to as Mean-Seeking behavior.
> The optimal decoder output $\bar{z}$ minimizing the MSE loss given a latent code $s$ is the conditional expectation ($\bar{z} = \mathbb{E}[z|s]$). When the encoder is noisy, the slot representation $s$ becomes ambiguous. Under this ambiguity, the decoder minimizes the error by averaging all plausible pixel realizations. This averaging inherently suppresses high-frequency details, resulting in a low-frequency, blurry approximation. Thus, encoder noise directly forces the decoded representation to be blurry.
>
> Regarding the claim that “Blurry Decoder Corrupts the Encoder’s Learning Signal”, we will tone down the phrase to “signal from blurry decoded output fails to provide the precise guidance needed to learn sharp object boundaries and refine the noisy patch representations”. This phenomenon occurs because sharp, high-frequency features are typically absent in the decoded representation for two primary reasons. First, due to the mean-seeking property of MSE [A], the decoder acts as a low-pass filter, failing to express high-frequency variations in the decoded output. This implies that the sensitivity of the decoder to high-frequency changes is negligible. In addition, according to [B], compressed features often lack precise spatial details, so that reconstructing from these compressed versions leads to blurry output.
> Consequently, since the decoder cannot reconstruct high-frequency details, the error signal derived from the reconstruction loss is inherently devoid of sharp, detailed information.
>
> Furthermore, to examine whether such a vicious cycle indeed arises during training, we conduct a qualitative analysis on the MOVi-C dataset by visualizing both attention maps and masks at the early and converged stages of training. The visualizations are provided in Appendix Fig. B4.
> In brief, the blurry decoded outputs in the early training stage lead multiple objects to collapse into a single slot during the training, and the attention noise observed early on is propagated into the decoder masks at convergence.
>
> A more detailed explanation of the qualitative analysis is provided in the revised version.
>
> [A] Image-to-Image Translation with Conditional Adversarial Networks. CVPR 2017.
>
> [B] Generating Images with Perceptual Similarity Metrics based on Deep Networks. NeurIPS 2016.

---

> ### Author Response · Authors · 2025-11-29
>
> ### **W6: No OCL results on image datasets. Why not testing on image datasets like COCO? Theoretically your method supports OCL on both images and videos.**
>
> We appreciate the reviewer's insight regarding the applicability of SRL to image domains. As the reviewer correctly pointed out, our Synergistic Representation Learning (SRL) is theoretically applicable to static images, as the "sharpness vs. smoothness" conflict is inherent to the Slot Attention architecture itself, independent of temporal dimensions. Following the suggestion, we evaluated SRL on the MS COCO dataset using the same training protocol as the baseline. The results are presented below:
>
> **Experimental results on COCO dataset**
> >| Method | ARI | mBO |
> >| --- | :---: | :---: |
> >| Recon only | 40.5 | 28.8 |
> >| Ours | 42.8 | 29.4 |
>
> As shown, SRL achieves a consistent improvement of +2.3 in ARI and +0.6 in mBO over the reconstruction-only baseline. This demonstrates that even without temporal cues, the mutual refinement between the encoder's sharp attention and the decoder's spatial coherence effectively improves object discovery. While our main paper focused on video benchmarks, these additional results confirm that SRL is a generalizable solution for object-centric learning.
>
> ### **W7: More hyperparameters, Line 341, i.e., $\lambda_{reg}$ and $\lambda_{CL}$, and top-K, as well as M at some where. Anyway, not a big deal. Most top conference fancy works would introduce one or two hyperparameters.**
>
> We acknowledge that our method introduces several hyperparameters such as loss weights ( $\lambda_{reg}$ and $\lambda_{CL}$ ), as well as K and M.
> Yet, we point out that M is simply fixed to a constant value (half of S) across all experiments, making it straightforward to set.
>
> Furthermore, to validate that our proposed SRL is not very sensitive to K, we have conducted experiments with unified K=16 across all datasets. Results are shown below, demonstrating that our SRL remains robust even under a unified configuration.
>
> **Experimental results with unified hyperparameters**
>
> >| | MOVI-C | | MOVI-E | |YTVIS2021 | |
> >| --- | :---: | :---: | :---: | :---: | :---: | :---: |
> >| | FG-ARI | MBO | FG-ARI | MBO | FG-ARI | MBO|
> >| Ours(maintable) |74.3 |34.5 |81.9|29.3|	42.9|35.6|
> >|Ours($K$=16)	|74.2|	34.9	|80.7|	29.7|	42.9|	35.6|
>
> ### **W8: More computation cost. According the two contrastive losses, pairwise cosine similarities need to be calculated and the three subsets need to be determined. So there must be most latency and VRAM consumption. It is necessary to provide quantitative results on these.**
>
> On the MOVi-C dataset, we measured the efficiency of SlotContrast and our SRL under the same setting (2×Quadro A6000 GPUs).
> In terms of VRAM per GPU, SlotContrast uses approximately 31.9 GB, whereas SRL uses 41.5 GB, corresponding to roughly a 30% increase in peak memory due to the additional contrastive objectives; the dominant cost from the DINOv2 backbone, slot attention, and decoder remains unchanged.
> For training time over 100,000 iterations, SlotContrast requires 28 h 27 m while SRL takes 31 h 21 m, which is about a 10% increase in wall-clock time.
> Additionally, we note that the inference time remains identical, as the contrastive losses are only used during training and do not introduce extra forward passes at test time.
> We believe this moderate overhead is justified by the consistent improvements in object discovery quality.
>
> ### **W9: Slot regularization contributes a large part to the total performance gain as shown in Tab 2. But this is not included in the main storyline.**
>
> We appreciate the reviewer pointing this out.
>
> Our intention was for slot regularization to be part of the main SRL story. It is designed to complement the encoder-decoder contrastive objectives by establishing a strong initial assignment of slots to objects, providing a reliable starting point for refining spatial maps based on both encoder and decoder signals. In particular, slot regularization prevents degenerate cases where multiple slots cover the same object; without slot regularization, the mutual refinement process might trigger the encoder and decoder to cooperate to further calibrate the semantic boundary of the fragmented object parts.
>
> We agree, however, that the current version under-emphasizes slot regularization in the main storyline. In the revision, we explained how it complements the encoder–decoder consistency signal and why it is necessary, both at the beginning of Section 3 and in Section 3.3.
>
> Also, we explicitly discuss its distinct role relative to existing slot-pruning approaches. This should make its connection to SRL’s overall design and motivation much clearer.

---

> ### Author Response · Authors · 2025-11-29
>
> ### **W10. Slot regularization is similar to slot pruning-related techniques, which can boost the performance almost in any cases. Thus related works should be compared or at least discussed:**
>
> [A] SOLV: Self-supervised Object-Centric Learning for Videos. NeurIPS 2023
> [B] MetaSlot: Break Through the Fixed Number of Slots in Object-Centric Learning. NeurIPS 2025.
>
> We appreciate the reviewer for pointing us to additional relevant works and fully agree that discussing them can help clarify the distinctions of our approach.
>
> SOLV [A] merges slots via an agglomerative clustering procedure, which is non-differentiable, while MetaSlot [B] addresses slot redundancy by using a codebook to prune duplicated slots.
> Although these approaches are effective at handling redundant slots, their mechanisms depend on redundancy being explicitly detected (e.g., clustering), which evolves slowly over training.
>
> We also employ a slot regularization objective to mitigate slot redundancy.
> However, unlike SOLV and MetaSlot, our slot regularization deliberately penalizes roughly half of the slots only during a very short initial phase, and then proceeds to the next stage to focus on slot representation learning. This explicitly encourages slots to separate more quickly, allowing the encoder and decoder to converge to stable spatial representations much earlier.
> As a result, our slot regularization offers an efficient way to establish a strong foundation on which SRL can subsequently operate.
>
> ### **W11: The proposed contrastive losses are actually a kind of fine-grained self-distillation, compared with the following existing works. Thus, related works should be compared or at least discussed:**
>
> [C] SPOT: Self-Training with Patch-Order Permutation for Object-Centric Learning with Autoregressive Transformers. CVPR 2024.
>     - offline "self"-distillation on slot attention masks
> [D] DIAS: Slot Attention with Re-Initialization and Self-Distillation. ACM MM 2025.
>     - online self-distillation on slot attention masks
> [E] SlotMatch: SlotMatch: Distilling Temporally Consistent Object-Centric Representations for Unsupervised Video Segmentation.
>     - offline "self"-distillation on slots
> [F] SmoothSA: Smoothing Slot Attention Iterations and Recurrences.
>
> We appreciate the reviewer for highlighting several related works on self-distillation, which helps clarify how our contrastive objectives differ from prior approaches.
>
> **Comparison with SPOT [C] and DIAS [D].**
>
> SPOT performs offline self-distillation by transferring the decoder’s attention patterns to the encoder, while DIAS performs online self-distillation by distilling later-iteration slot attention into earlier iterations. Although these methods are effective in certain settings, both approaches directly mimic the teacher's attention signals, which may introduce noisy signals.
> In contrast, our contrastive objectives are designed to leverage only the complementary strengths of encoder and decoder representations.
> We mitigate the impact of noisy signals by stratifying mutual signals into intermediate levels, rather than enforcing strong positive or negative constraints.
> These methods are updated in our related work section.
>
> **Comparison with SlotMatch [E] and SmoothSA [F]**
>
> First, we note that SlotMatch and SmoothSA are concurrent works. Nonetheless, to provide a comparison, these methods adopt MSE or KLD-based distillation losses, which primarily accelerate optimization by encouraging consistency with the distillation source. However, they still lack specific mechanisms to mitigate the noise inherent in the teacher signals, which limits their robustness in unsupervised scenarios. In contrast, our method employs a contrastive formulation that inherently downweights ambiguous or noisy pairs while emphasizing reliable positive relations, enabling more stable and noise-resistant representation learning.

---

### Official Review · Reviewer_7nVy · 2025-10-31

**Soundness:** 2
**Presentation:** 3
**Contribution:** 2
**Rating:** 4
**Confidence:** 3

**Summary:**

The paper modifies SlotContrast, a framework for extracting object-centric representations from pretrained vision foundation models, to produce sharper and less noisy outputs for each slot. The proposed approach consists of three stages: in the first stage (the initial 10% of training), slot–patch correspondence maps are extracted, based on which sets of positive and negative pairs are formed. The decoder loss is then regularized with a contrastive loss derived from these sets. In the second stage (10–20% of training), no regularization is applied. Finally, after 20% of training, the encoder features are regularized, but the sets are now extracted from the decoder’s object maps. Empirical results demonstrate improvements over the baseline, yielding sharper and cleaner maps.

**Strengths:**

1 - The problem discussed in the paper is an interesting observation, which has not been explored in the previous works.

2 - The paper shows better empirical results than state-of-the-art.

3 - The paper is clean, and the method has been elaborated in details for different stages.

**Weaknesses:**

1 - Although I agree with the identified problem, the proposed solution appears overly complex. It consists of three stages defined by the proportion of training iterations completed. Since different backbones or even datasets may require varying numbers of iterations, the approach seems unlikely to generalize well. Could you please report results with other foundation models such as Dino3 [1] or Franca [2] to verify generalizability?

2 - Recently, several post-training methods have been introduced to address the issue of noisy dense features in foundation models [3, 4]. A simpler alternative might be to replace the backbone with such improved checkpoints and apply SlotContrast directly. This approach could provide a more natural and straightforward solution than introducing complex regularization. Could you also compare performance using these backbones?

3 - An ablation study on the parameter η, which determines the stage at which each loss function is applied, would be valuable. This would further demonstrate the robustness of the method under different training schemes.

1 - Dinov3." arXiv preprint arXiv (2025)

2 - Franca: Nested Matryoshka Clustering for Scalable Visual Representation Learning, arXiv preprint arXiv (2025)

3 - MoSiC: Optimal-Transport Motion Trajectory for Dense Self-Supervised Learning, ICCV25

4 - Near, far: Patch-ordering enhances vision foundation models' scene understanding, ICLR25

**Questions:**

I explained my question in the weaknesses, please refer to them.

---

> ### Author Response · Authors · 2025-11-29
>
> ### **W1: It consists of three stages defined by the proportion of training iterations completed. Since different backbones or even datasets may require varying numbers of iterations, the approach seems unlikely to generalize well. Could you please report results with other foundation models such as Dino3 [1] or Franca [2] to verify generalizability?**
>
> To verify the generalizability of SRL, we have conducted experiments using Franca ViT-B/16 below.
>
> We used the same training and evaluation protocol as reported in the manuscript.
>
> **Experimental results with Franca backbone**
> >|  | MOVi-C |  | YouTube-VIS |  |
> >| --- | :---: | :---: | :---: | :---: |
> >| Method | FG-ARI | mBO | FG-ARI | mBO |
> >| SlotContrast | 66.8 | 35.6 | 35.3 | 32.7 |
> >| Ours | 66.1 | 37.2 | 38.9 | 36.4 |
>
> When using Franca as the backbone, SRL is particularly beneficial on the mBO metric on MOVi-C, and surpasses SlotContrast by a large margin on YTVIS 2021.
> These results confirm that SRL is not only effective with the DINO ViT backbone but also generalizable to other types of backbones.
>
> These results are updated in the Appendix.
>
> ### **W2: Recently, several post-training methods have been introduced to address the issue of noisy dense features in foundation models. A simpler alternative might be to replace the backbone with such improved checkpoints and apply SlotContrast directly. This approach could provide a more natural and straightforward solution than introducing complex regularization. Could you also compare performance using these backbones?**
>
> We thank the reviewer for pointing out recent post-training methods (i.e., MoSiC) and the suggestion to simply replace the backbone with such improved checkpoints. To investigate this, we replaced DINOv2 with a MoSiC-s14 backbone and ran SlotContrast and our method under the same protocol. The results are:
>
> **Experimental results with MoSiC backbone**
> >| Method | FG-ARI | mBO |
> >| --- | --- | --- |
> >| DINOv2-s14 SlotContrast | 70.4 | 31.7 |
> >| MoSiC-s14 SlotContrast | 70.3 | 31.6 |
> >| MoSiC-s14 Ours | 74.3 | 37.2 |
>
> This result shows that SRL provides consistent gains even when built on a denoised backbone such as MoSiC. In fact, we observe that such denoising may introduce new types of noisy patches. For example, MoSiC features often exhibit a stronger spatial/positional bias: in some videos, slots cluster over empty background regions or fail to track moving objects, suggesting that part of the denoising comes at the cost of new artifacts (qualitative results have been added to the Appendix). Nevertheless, SRL effectively mitigates these artifacts and restores meaningful object assignments, yielding a substantial performance improvement over SlotContrast.
>
> These experimental results and the qualitative analysis are included in the Appendix.

---

> ### Author Response · Authors · 2025-11-29
>
> ### **W3: An ablation study on the parameter η, which determines the stage at which each loss function is applied, would be valuable. This would further demonstrate the robustness of the method under different training schemes.**
>
> We thank the reviewer for suggesting an ablation on the schedule parameter $\eta$, which controls when each loss is active. In our method, we use staged training with (i) a slot regularization loss applied only in the early phase, and (ii) a feature-learning (denoising/deblurring) loss that is turned on later in training. Below, we report ablations on both stage boundaries, measured on our two main metrics (FG-ARI / mBO).
>
> **(a) When to stop slot regularization**
>
> We vary the iteration at which slot regularization is turned off, while keeping the rest of the training protocol fixed:
>
> **Table: Effect of slot-regularization ending iteration**
>
> >| End iter (slot reg) | FG-ARI | mBO |
> >| --- | :---: | :---: |
> >| 5k | 72.07 | 34.96 |
> >| 10k (default in paper) | 74.30 | 34.50 |
> >| 20k | 73.96 | 35.05 |
>
> Recall that the SlotContrast baseline in this setting is (70.8 / 31.4). We observe that all schedules substantially outperform the baseline, and performance varies smoothly. Later stopping (20k) yields slightly better mBO, indicating that the method does not rely on a very specific early stopping point.
>
> **(b) When to start the feature-learning loss**
>
> We also vary the iteration at which contrastive learning objectives are activated:
>
> **Table: Effect of contrastive learning objectives starting iteration**
>
> >| Start iter (feat learn) | FG-ARI | mBO |
> >| --- | :---: | :---: |
> >| 0 | 71.45 | 34.04 |
> >| 10k | 71.77 | 32.76 |
> >| 20k (default in paper) | 74.30 | 34.50 |
> >| 25k | 75.65 | 33.11 |
> >| 30k | 73.61 | 34.12 |
> >| 40k | 75.14 | 35.40 |
>
> Again, all configurations are clearly above the SlotContrast baseline, and the performance changes gradually as we shift the starting point earlier or later. The best values are obtained when contrastive objectives are activated at mid-to-late stages (20k–40k), but the range of decent settings is wide.
>
> These ablations show that our method is not very sensitive to the choice of $\eta$, as SRL consistently improves over SlotContrast for all tested schedules. Also, we point out that performance varies smoothly rather than collapsing outside a narrow configuration.
>
> These results are added in the Appendix.

---

### Official Review · Reviewer_o8sx · 2025-11-01

**Soundness:** 2
**Presentation:** 3
**Contribution:** 2
**Rating:** 4
**Confidence:** 3

**Summary:**

This paper introduces Synergistic Representation Learning (SRL), a method designed to resolve the conflict between the sharp encoder attention maps and the blurry decoder reconstructions in unsupervised object-centric learning. SRL enables mutual refinement between the encoder and decoder, leveraging the encoder's sharpness to enhance the decoder's output and the decoder's spatial consistency to improve the encoder's features. The method is validated on three video object-centric learning benchmarks.

**Strengths:**

1. The paper presents a novel opinion: a vicious cycle between the encoder and decoder in video object-centric learning.
2. The paper is well-organized, and the experimental design is clear.

**Weaknesses:**

1. The authors propose a vicious cycle in unsupervised video object-centric learning, where noisy encoder inputs lead to blurry, low-frequency decoder outputs, which in turn fail to refine the encoder's features. However, it remains unclear whether this phenomenon truly exists during training, and whether it worsens as training progresses. Qualitative or quantitative experiments are necessary to justify this claim.
2. In the comparison experiments presented in Table.1, SRL does not show a clear advantage across multiple metrics (with three out of six metrics showing actually lower performance). It would be beneficial to evaluate the method's effectiveness on additional datasets.
3. The implementation details and ablation studies suggest that the model's performance on a benchmark is somewhat sensitive to hyperparameters, such as the number of positive patches (K) and the number of slots. This raises a question: whether the proposed method is task-specific.
4. How does the proposed method's efficiency (in terms of both time and space cost) compare to previous methods?
5. From the visualization results, both the proposed method and the comparison methods seem to produce relatively coarse object discovery and segmentation results, with limited delineation of object boundaries. Thus, what advantages do these methods offer over models specifically designed for object or instance segmentation in real-world scenarios?

**Questions:**

Please see the weaknesses.

---

> ### Author Response · Authors · 2025-11-29
>
> ### **W1: The authors propose a vicious cycle in unsupervised video object-centric learning, where noisy encoder inputs lead to blurry, low-frequency decoder outputs, which in turn fail to refine the encoder's features. However, it remains unclear whether this phenomenon truly exists during training, and whether it worsens as training progresses. Qualitative or quantitative experiments are necessary to justify this claim.**
>
> To investigate whether the vicious cycle between attention noise and mask blur indeed arises during training, we conduct a qualitative analysis on the MOVi-C dataset by visualizing both attention maps and masks at the early and converged stages of training. The visualizations are provided in Appendix Fig. B4.
>
> To briefly illustrate, we observe that the quality of both encoder attention maps and decoder masks can deteriorate as training progresses for SlotContrast.
> Some objects that are initially well separated may gradually lose their semantic boundaries, causing multiple objects to be merged into a single region.
> Also, blurred boundaries prevent the model from disentangling overlapping objects, and residual attention noise persists even after training, propagating into the decoder masks.
>
> In contrast, our method effectively suppresses this error propagation.
> Even when some objects are under-segmented at early stages, the deblurring process of semantic boundaries encourages the model to recover clear object-wise separation as training proceeds.
> In addition, although the encoder attention maps initially exhibit noisy and blurred boundaries, our method progressively removes this noise and yields sharper encoder attention and cleaner decoder masks by the end of training.
>
> - - -
>
> ### **W2: In the comparison experiments presented in Tab.1, SRL does not show a clear advantage across multiple metrics (with three out of six metrics showing actually lower performance). It would be beneficial to evaluate the method's effectiveness on additional datasets.**
>
> We thank the reviewer for pointing out that Table 1 alone may not fully convey the advantages of SRL, and for suggesting evaluation on additional datasets.
>
> To test whether SRL transfers beyond the datasets used in the main paper, we follow the VideoSAUR protocol and evaluate on DAVIS using a model trained on YTVIS-2021. We use a 37×37 patch grid and report the standard DAVIS metrics J, F, and J&F, following VideoSAUR.
>
> **Results on DAVIS**
>
> >| Method | F | J | J&F |
> >|---|:---:|:---:|:---:|
> >| SlotContrast | 22.2 | 36.5 | 29.3 |
> >| Ours | 25.4 | 48.2 | 36.8 |
>
> SRL improves J by +11.7 points and J&F by +7.5 points, indicating substantially better object discovery quality under this widely used protocol.
>
> Furthermore, we evaluated on YTVIS-2019 under two settings: (1) transfer from YTVIS-2021 and (2) train on the YTVIS-2019 train set and evaluate on its validation set.
>
> In both settings, SRL improves over SlotContrast on ARI and mBO as reported below.
>
> **Results on YTVIS19 (Transferred from YTVIS-2021)**
>
> >| Method | FG-ARI | mBO |
> >| --- | :---: | :---: |
> >| SlotContrast | 16.6 | 43.3 |
> >| Ours | 20.4 | 53.3 |
>
> **Results on YTVIS19 (Trained on YTVIS19 train set)**
>
> >| Method | FG-ARI | mBO |
> >|---|:---:|:---:|
> >| SlotContrast | 16.7 | 44.9 |
> >| Ours | 19.1 | 46.9 |
>
> Details of the experiment settings are elaborated in the revised manuscript.

---

> ### Author Response · Authors · 2025-11-29
>
> ### **W3: The implementation details and ablation studies suggest that the model's performance on a benchmark is somewhat sensitive to hyperparameters, such as the number of positive patches (K) and the number of slots. This raises a question: whether the proposed method is task-specific.**
>
> We agree that the number of slots and the number of positive patches (K) are important hyperparameters in slot-based VOCL. However, we do not observe that SRL is unusually sensitive or task-specific.
>
> First, for the number of slots, we followed our baseline SlotContrast for a fair comparison.
>
> Also, for the number of positive patches (K) and the number of slots to be penalized for regularization (M), we illustrated that all results with varying K and M yield performances that clearly outperform the SlotContrast baseline (70.8 / 31.4 on the corresponding metrics), rather than peaking at a single, narrow configuration, as shown in Fig. 3. In other words, while absolute performance naturally varies with K and M (as is typical in unsupervised object-centric learning), the gains from SRL are stable and consistent over this range.
>
> Second, because the training is fully unsupervised, it is a standard practice (also in prior work [A]) to choose a reasonable number of positives for contrastive learning strategies via simple thresholding or top-K selection. Our goal with the ablations was to be transparent about this design choice, not to imply brittleness. Nevertheless, we demonstrate that our proposed SRL works well even with the unified K across all datasets, as reported below.
>
> **Evaluation with the unified hyperparameters**
> >|  | MOVI-C |  | MOVI-E |  | YTVIS2021 |  |
> >| --- | :---: | :---: | :---: | :---: | :---: | :---: |
> >|  | FG-ARI | MBO | FG-ARI | MBO | FG-ARI | MBO |
> >| Ours(maintable) | 74.3 | 34.5 | 81.9 | 29.3 | 42.9 | 35.6 |
> >| Ours($K$=16) | 74.2 | 34.9 | 80.7 | 29.7 | 42.9 | 35.6 |
>
> [A] Seong et al. Leveraging Hidden Positives for Unsupervised Semantic Segmentation, CVPR 2023
>
> ### **W4: How does the proposed method's efficiency (in terms of both time and space cost) compare to previous methods?**
>
> On MOVi-C, we measured the efficiency of SlotContrast and our SRL under the same setting (2×Quadro A6000 GPUs). In terms of VRAM per GPU, SlotContrast uses approximately 31.9 GB, whereas SRL uses 41.5 GB, corresponding to roughly a 30% increase in peak memory due to the additional contrastive objectives; the dominant cost from the DINOv2 backbone, slot attention, and decoder remains unchanged. For training time, SlotContrast requires 28 h 27 m while SRL takes 31 h 21 m, i.e., about a 10% increase in wall-clock training time. Importantly, inference time is identical, since the contrastive losses are used only during training and do not introduce extra forward passes at test time. We believe this moderate overhead is justified by the consistent improvements in object discovery quality and cross-dataset robustness.

---

> ### Author Response · Authors · 2025-11-29
>
> ### **W5: From the visualization results, both the proposed method and the comparison methods seem to produce relatively coarse object discovery and segmentation results, with limited delineation of object boundaries. Thus, what advantages do these methods offer over models specifically designed for object or instance segmentation in real-world scenarios?**
>
> We agree that both our method and other unsupervised object-centric approaches produce relatively coarse boundaries when visually compared to instance-segmentation methods. However, our goal is fundamentally different from that of models for object or instance segmentation.
>
> First, we point out that our method operates in the fully unsupervised object-centric learning setting: it uses no ground-truth masks or class labels, and is trained on large collections of unlabeled videos. In contrast, state-of-the-art instance segmentation models rely on dense human annotations for each frame and for each object category (even the unsupervised VIS methods rely on ground-truth masks during training). In many realistic scenarios we target (long, uncurated videos, new domains, or proprietary data), such dense labels are either unavailable or prohibitively expensive to obtain. SRL is designed for these label-scarce settings.
>
> Also, the final outputs of object-centric learning differ from those of segmentation models. The primary output of our model is a set of slot representations that decomposes videos into disentangled, object-level latent vectors that can be directly consumed by downstream modules (e.g., for object dynamics prediction as reported below). Per-pixel masks are only a diagnostic visualization of these slots, not the main product of the method.
>
> **Object Dynamics Prediction Task**
>
> To test whether our method benefits downstream tasks, we evaluate our pretrained video object-centric models on an object dynamics prediction task. Following SlotContrast, we attach a dynamics module on top of the frozen object-centric encoder and train it to predict future slots. We use the identical experimental setup introduced in SlotContrast, ensuring a fair comparison. Compared to the baseline (trained only with the reconstruction objective) and SlotContrast, SRL produces slot representations that are effective in downstream applications.
>
> A detailed experiment settings are elaborated in the revised version.
>
> >|  | MOVi-C |  | YTVIS2021 |  | MOVi-E |  |
> >| --- | :---: | :---: | :---: | :---: | :---: | :---: |
> >| Method | FG-ARI | mBO | FG-ARI | mBO | FG-ARI | mBO |
> >| Baseline | 50.7 | 25.9 | 27.4 | 28.9 | 70.6 | 24.3 |
> >| SlotContrast | 63.8 | 26.1 | 29.2 | 29.6 | 70.5 | 24.9 |
> >| Ours | 68.9 | 27.4 | 32.2 | 30.0 | 70.4 | 24.9 |

---

### Author Response · Authors · 2025-12-01
**Summary of the rebuttal**

Dear ACs, SACs, and PCs,

We sincerely appreciate the continued effort and attention ensuring a fair evaluation process, despite this extraordinary review cycle.

To assist with the evaluation, we have compiled a summary of the reviewers’ main concerns along with our corresponding responses. This overview highlights the essential experiments, insights, and modifications that strengthen the SRL framework.

---
---

**1. Validating the Existence of the Vicious Cycle** ```[o8sx, n3E7, 18N8]```

>- **Theoretical**: We have grounded the explanation in known properties of MSE reconstruction (mean-seeking, low-pass behavior) and of reconstruction from compressed features.
>- **Empirical**: We also provided a new qualitative analysis tracking attention and mask evolution in the revised manuscript.  **(Appendix B.6)**

**2. Backbone Dependence & Dataset Generalizability** ```[7nVy, o8sx, 18N8]```

>We confirmed robust generalizability through extensive new experiments.
> - **Backbone Agnostic**: For new backbones, we replaced DINOv2 with Franca and MoSiC **(Appendix B.3)**
>    + Compared to SlotContrast, SRL exhibit +3.6%p ARI, +3.7%p mBO for Franca backbone (YTVIS2021 dataset) | +4%p ARI, +5.6%p mBO for MoSiC backbone (MOVi-C dataset)
>- **Dataset Scalability**: For new datasets, we evaluated on DAVIS (transfer setting), YTVIS-2019, and MS COCO (image domain). **(Appendix B.2, Appendix B.4)**
>    + Compared to SlotContrast, SRL exhibit +7.5%p J&F on DAVIS | +3.8%p ARI, +10%p mBO on YTVIS19 | +2.3%p ARI, +0.6%p mBO on COCO
>- **Downstream Task**: Additional task: we have applied our learned slots for object dynamics prediction task. **(Section 4.3)**
>    + Compared to SlotContrast, SRL exhibit +5.1%p ARI, +1.3%p mBO on MOVI-C dataset

**3. Robustness to Hyperparameters (e.g., $K$, $M$, $\eta$)** ```[o8sx, 7nVy]```

>We conducted comprehensive ablation studies.
>- **Stability**: SRL maintains SOTA performance even when using a fixed hyperparameters across all datasets, removing the need for dataset-specific tuning
>- **Schedule Robustness**: We have conducted extensive ablations on the start/end iterations of the loss terms which showed smooth performance curves, indicating the method is stable and not brittle to schedule changes  **(Appendix B.5)**

**4. Efficiency and overhead** ```[18N8, o8sx]```

> We measured the efficiency of SlotContrast and our SRL under the same setting.
> - **Training time**: Our SRL requires takes a 10% increase in wall-clock training time
> - **VRAM**: Our SRL uses roughly a 30% increase in peak memory due to the additional contrastive objectives; the dominant cost from the DINOv2 backbone, slot attention, and decoder remains unchanged
> - **Inference time** is identical, since the contrastive losses are used only during training
> - We believe this moderate overhead is justified by the consistent improvements in object discovery quality and cross-dataset robustness

**5. Reconstruction Loss Variants (MSE vs. MAE)** ```[n3E7]```

> Reviewer asked if the blurriness is simply due to MSE and if MAE would solve it.
>- **Loss Agnostic**: While MAE reduces some blur, it introduces new noise and under-segmentation issues. SRL was shown to improve performance significantly even under the MAE setting (enhance 1.4%p ARI and 6.8%p mBO over SlotContrast) **(Appendix B.1)**

**6. Decoder Variants (MLP vs Transformer)** ```[n3E7]```

> Reviewer asked if the poor reconstruction maps are a result of the choice of MLP, and the other types of decoders suffer the same issue.
>- MLP decoders are the standard choice in object-centric learning, and addressing their inherent blurriness is of high practical importance. We therefore use MLP decoders to remain consistent with prior work and to isolate the effect of SRL
>- We additionally tested Transformer decoders. They produce sharper boundaries but often introduce over-fragmentation (lower FG-ARI), poor scalability to high resolutions, and require heavy tuning for each dataset, consistent with both prior reports and our observations
>- Applying SRL to Transformer decoder still improves performance; although the relative improvement is slightly attenuated compared to the MLP setting (likely because the baseline blurriness is less severe), the benefits of our mutual refinement framework are evident

**7. Related Work** ```[18N8]``` **and Failure Cases** ```[n3E7]```
>We have elaborated on the comparison with additional related works (SOLV, MetaSlot, SPOT, DIAS) and the failure cases in the revised manuscript. **(Section 2 and Appendix A.2)**

---
---

The revised paper has been uploaded, and we provided detailed rebuttals to the reviewers’ extensive and high-quality comments below. We believe the added clarifications and experiments substantially strengthen both the justification and overall quality of our work. We kindly request a careful consideration in light of these improvements.

Sincerely,

Authors

---

### Meta-Review · Area_Chair_BVPm · 2026-01-05

**Summary:**

This paper builds on the recent SlotContrast method, a method for unsupervised object discovery/segmentation and tracking in video data, and introduces a refinement method to synergistically refine encoder features / decoder sharpness. Results demonstrate clear gains in typical video object discovery metrics.

The initial reviewer reception was borderline, but the authors did a great job in addressing most concerns, which would have likely convinced most if not all reviewers to upgrade their score to weak accept or higher.

Overall, even though this did not come up during the review/discussion phase, it would have been great to compare this method to diffusion-based approaches such as SlotDiffusion (Wu et al., NeurIPS 2023) which share a similar motivation in overcoming limitations of deterministic decoders and their associated blurry reconstructions. Nonetheless, the paper should be of interest to the community, is of high quality, has sufficient novelty, and hence can be accepted. The authors are encouraged to contextualize their method against diffusion-based approaches in the final version of the paper.

**Reviewer Concerns:**

All concerns relating to experimental validation and related works have been sufficiently addressed.

Remaining concerns are the complexity of the method (building on the already complex SlotContrast method) and the soundness of the motivation. The authors slightly revised their motivation statement in their rebuttal, which alleviates this concern.

**Reviewer Scores:**

All reviewers initially had a similar borderline assessment of the paper and most if not all would have likely increased their score after the successful rebuttal by the authors.

---

### Decision · Program_Chairs · 2026-01-26

Accept (Poster)